# Overtone focusing in biphonic tuvan throat singing

**Christopher Bergevin[1,2,3,4]\*, Chandan Narayan[5], Joy Williams[6], Natasha Mhatre[7], Jennifer KE Steeves[2,8], Joshua GW Bernstein[9], Brad Story[10]\***

[1]Physics and Astronomy, York University, Toronto, Canada; [2]Centre for Vision Research, York University, Toronto, Canada; [3]Fields Institute for Research in Mathematical Sciences, Toronto, Canada; [4]Kavli Institute of Theoretical Physics, University of California, Santa Barbara, United States; [5]Languages, Literatures and Linguistics, York University, Toronto, Canada; [6]York MRI Facility, York University, Toronto, Canada; [7]Biology, Western University, London, Canada; [8]Psychology, York University, Toronto, Canada; [9]National Military Audiology & Speech Pathology Center, Walter Reed National Military Medical Center, Bethesda, United States; [10]Speech, Language, and Hearing Sciences, University of Arizona, Tucson, United States

**Abstract** Khoomei is a unique singing style originating from the republic of Tuva in central Asia. Singers produce two pitches simultaneously: a booming low-frequency rumble alongside a hovering high-pitched whistle-like tone. The biomechanics of this biphonation are not well-understood. Here, we use sound analysis, dynamic magnetic resonance imaging, and vocal tract modeling to demonstrate how biphonation is achieved by modulating vocal tract morphology. Tuvan singers show remarkable control in shaping their vocal tract to narrowly focus the harmonics (or overtones) emanating from their vocal cords. The biphonic sound is a combination of the fundamental pitch and a focused filter state, which is at the higher pitch (1–2 kHz) and formed by merging two formants, thereby greatly enhancing sound-production in a very narrow frequency range. Most importantly, we demonstrate that this biphonation is a phenomenon arising from linear filtering rather than from a nonlinear source.

**\*For correspondence:**
cberge@yorku.ca (CB);
bstory@email.arizona.edu (BS)

**Competing interests:** The authors declare that no competing interests exist.

## Introduction

In the years preceding his death, Richard Feynman had been attempting to visit the small republic of Tuva located in geographic center of Asia (*Leighton, 2000*). A key catalyst came from Kip Thorne, who had gifted him a record called *Melody tuvy*, featuring a Tuvan singing in a style known as Khoomei, or Xöömij. Although he was never successful in visiting Tuva, Feynman was nonetheless captivated by Khoomei, which can be best described as a high-pitched tone, similar to a whistle carrying a melody, hovering above a constant booming low-frequency rumble. This is a form of biphonation, or in Feynman's own words, "a man with two voices". Khoomei, now a part of the UNESCO Intangible Cultural Heritage of Humanity, is characterized as "the simultaneous performance by one singer of a held pitch in the lower register and a melody . . . in the higher register" (*Aksenov, 1973*). How, indeed, does one singer produce two pitches at one time? Even today, the biophysical underpinnings of this biphonic human vocal style are not fully understood.

Normally, when a singer voices a song or speech, their vocal folds vibrate at a fundamental frequency ($f_0$), generating oscillating airflow, forming the so-called *source*. This vibration is not, however, simply sinusoidal, as it also produces a series of harmonics tones (i.e., integer multiples of $f_0$) (**Figure 1**). Harmonic frequencies in this sound above $f_0$ are called overtones. Upon

**eLife digest** The republic of Tuva, a remote territory in southern Russia located on the border with Mongolia, is perhaps best known for its vast mountainous geography and the unique cultural practice of "throat singing". These singers simultaneously create two different pitches: a low-pitched drone, along with a hovering whistle above it. This practice has deep cultural roots and has now been shared more broadly via world music performances and the 1999 documentary Genghis Blues.

Despite many scientists being fascinated by throat singing, it was unclear precisely how throat singers could create two unique pitches. Singing and speaking in general involves making sounds by vibrating the vocal cords found deep in the throat, and then shaping those sounds with the tongue, teeth and lips as they move up the vocal tract and out of the body. Previous studies using static images taken with magnetic resonance imaging (MRI) suggested how Tuvan singers might produce the two pitches, but a mechanistic understanding of throat singing was far from complete.

Now, Bergevin et al. have better pinpointed how throat singers can produce their unique sound. The analysis involved high quality audio recordings of three Tuvan singers and dynamic MRI recordings of the movements of one of those singers. The images showed changes in the singer's vocal tract as they sang inside an MRI scanner, providing key information needed to create a computer model of the process.

This approach revealed that Tuvan singers can create two pitches simultaneously by forming precise constrictions in their vocal tract. One key constriction occurs when tip of the tongue nearly touches a ridge on the roof of the mouth, and a second constriction is formed by the base of the tongue. The computer model helped explain that these two constrictions produce the distinctive sounds of throat singing by selectively amplifying a narrow set of high frequency notes that are made by the vocal cords. Together these discoveries show how very small, targeted movements of the tongue can produce distinctive sounds.

emanating from the vocal folds, they are then sculpted by the vocal tract, which acts as a spectral *filter*. The vocal-tract filter has multiple resonances that accentuate certain clusters of overtones, creating *formants*. When speaking, we change the shape of our vocal tract to shift formants in systematic ways characteristic of vowel and consonant sounds. Indeed, singing largely uses vowel-like sounds (*Story, 2016*). In most singing, the listener perceives only a single pitch associated with the $f_0$ of the vocal production, with the formant resonances determining the timbre. Khoomei has two strongly emphasized pitches: a low-pitch drone associated with the $f_0$, plus a melody carried by variation in the higher frequency formant that can change independently (*Kob, 2004*). Two possible loci for this biphonic property are the *source* and/or the *filter*.

A source-based explanation could involve different mechanisms, such as two vibrating nonlinear sound sources in the syrinx of birds, which produce multiple notes that are harmonically unrelated (*Fee et al., 1998*; *Zollinger et al., 2008*). Humans however are generally considered to have only a single source, the vocal folds. But there are an alternative possibilities: for instance, the source could be nonlinear and produce harmonically-unrelated sounds. For example, aerodynamic instabilities are known to produce biphonation (*Mahrt et al., 2016*). Further, Khoomei often involves dramatic and sudden transitions from simple tonal singing to biophonation (see *Figure 1* and the Appendix for associated audio samples). Such abrupt changes are often considered hallmarks of physiological nonlinearity (*Goldberger et al., 2002*), and vocal production can generally be nonlinear in nature (*Herzel and Reuter, 1996*; *Mergell and Herzel, 1997*; *Fitch et al., 2002*; *Suthers et al., 2006*). Therefore it remains possible that biphonation arises from nonlinear source considerations.

Vocal tract shaping, a filter-based framework, provides an alternative explanation for biphonation. In one seminal study of Tuvan throat singing, Levin and Edgerton examined a wide variety of song types and suggested that there were three components at play. The first two ('tuning a harmonic' relative to the filter and lengthening the closed phase of the vocal fold vibration) represented a coupling between source and filter. But it was the third, narrowing of the formant, that appeared crucial. Yet, the authors offered little empirical justification for how these effects are produced by the vocal tract shape in the presented radiographs. Thus it remains unclear how the high-pitched formant in

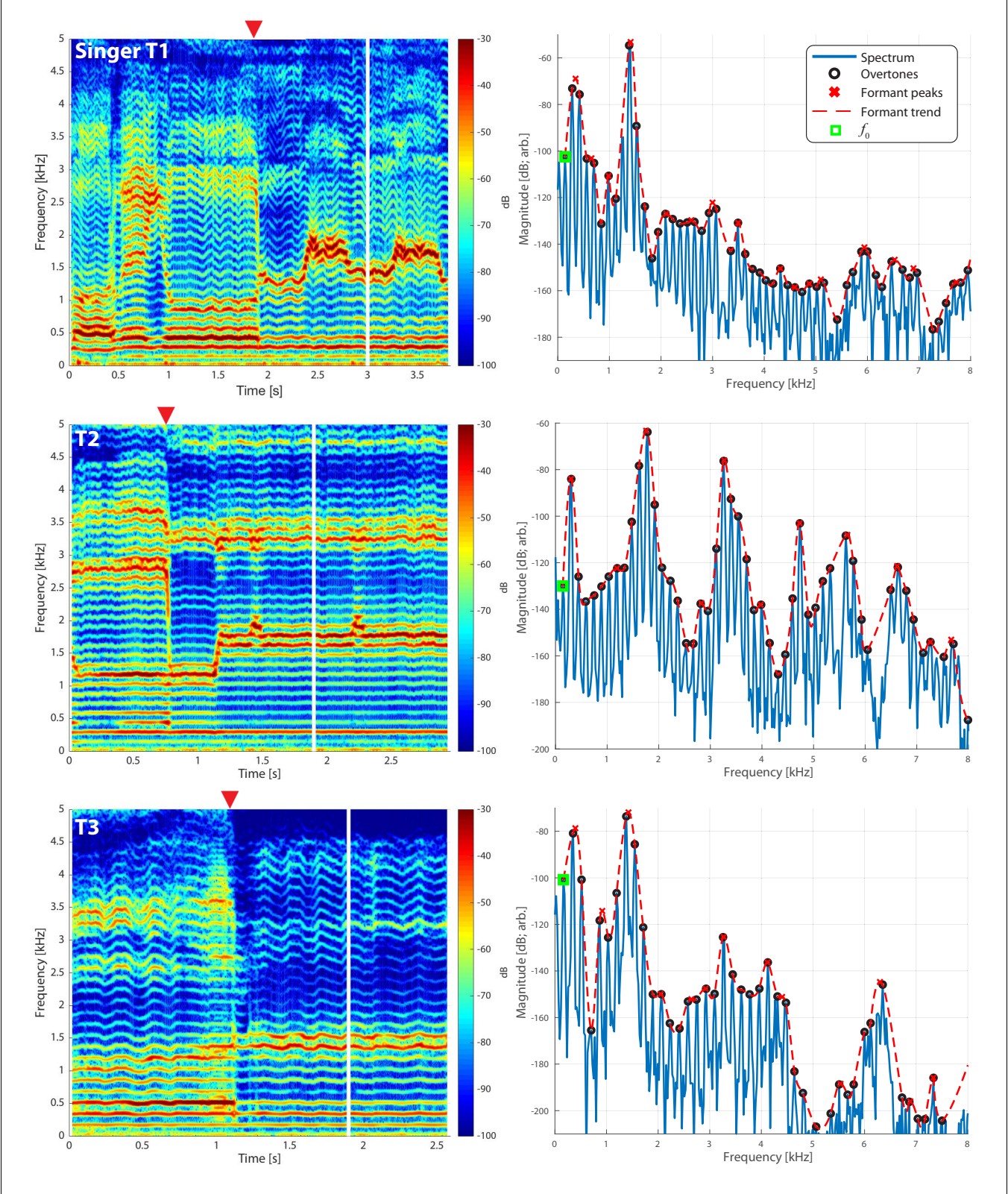

**Figure 1.** Frequency spectra for three different singers transitioning from normal to biphonic singing. Vertical white lines in the spectrograms (left column) indicate the time point for the associated spectrum in the right column. Transition points from normal to biphonic singing state are denoted by the red triangle. The fundamental frequency ($f_0$) of the song is indicated by a peak in the spectrum marked by a green square. Overtones, which represent integral multiples of this frequency, are also indicated (black circles). Estimates of the formant structure are shown by overlaying a red dashed

*Figure 1 continued on next page*

*Figure 1 continued*

line and each formant peak is marked by an x. Note that the vertical scale is in decibels (e.g., a 120 dB difference is a million-fold difference in pressure amplitude). See also *Appendix 1—figure 1* and *Appendix 1—figure 2* for further quantification of these waveforms. The associated waveforms can be accessed in the Appendix [T1_3short.wav, T2_5short.wav, T3_2shortA.wav].

Khoomei was formed (*Grawunder, 2009*). Another study (*Adachi and Yamada, 1999*) examined a throat singer using magnetic resonance imaging (MRI) and captured static images of the vocal tract shape during singing. These images were then used in a computational model to produce synthesized song. Adachi and Yamada argued that a "rear cavity" was formed in the vocal tract and its resonance was essential to biphonation. However, their MRI data reveal limited detail since they were static images of singers already in the biphonation state. Small variations in vocal tract geometry can have pronounced effects on produced song (*Story et al., 1996*) and data from static MRI would reveal little about how and which parts of the vocal tract change shape as the singers transition from simple tonal song to biphonation. To understand which features of vocal tract morphology are crucial to biophonation, a dynamic description of vocal tract morphology would be required.

Here we study the dynamic changes in the vocal tracts of multiple expert practitioners from Tuva as they produce Khoomei. We use MRI to acquire volumetric 3D shape of the vocal tract of a singer during biphonation. Then, we capture the dynamic changes in a midsagittal slice of the vocal tract as singers transition from tonal to biphonic singing while making simultaneous audio recordings of the song. We use these empirical data to guide our use of a computational model, which allows us to gain insight into which features of vocal tract morphology are responsible for the singing phonetics observed during biophonic Khoomei song (e.g., *Story, 2016*). We focus specifically on the Sygyt (or Sigit) style of Khoomei (*Aksenov, 1973*).

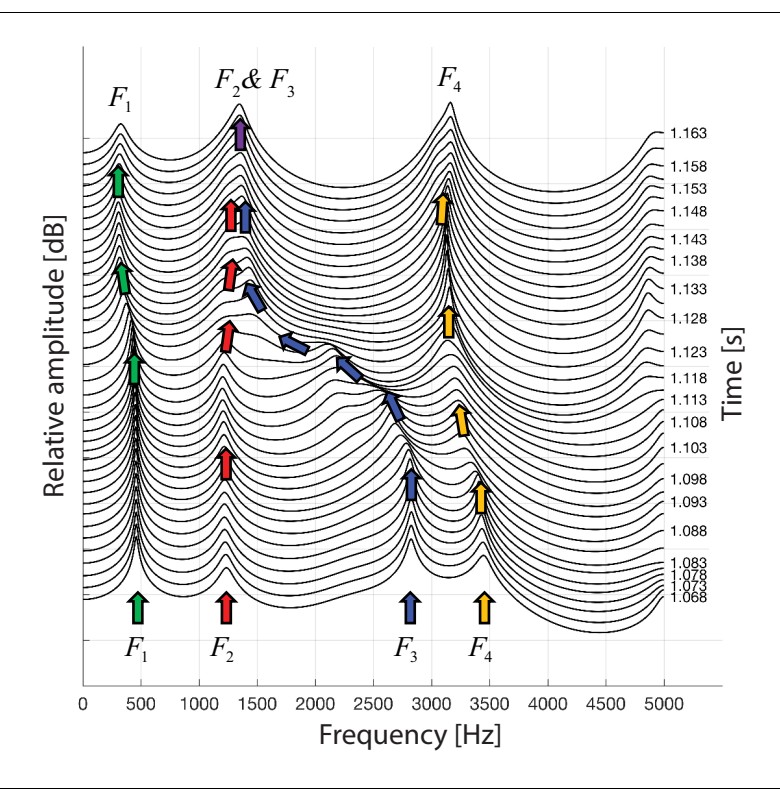

**Figure 2.** A waterfall plot representing the spectra at different time points as singer T2 transitions from normal singing into biphonation (T2_3short.wav). The superimposed arrows are color-coded to help visualize how the formants change about the transition, chiefly with F3 shifting to merge with F2. This plot also indicates the second focused state centered just above 3 kHz is a sharpened F4 formant.

## Results

### Audio recordings

We made measurements from three Tuvan singers performing Khoomei in the Sygyt style (designated as T1–T3) and one (T4) in a non-Sygyt style. Songs were analyzed using short-time Fourier transforms (STFT), which provide detailed information in both temporal and spectral domains. We recorded the singers transitioning from normal singing into biphonation, *Figure 1* showing this transition for three singers. The $f_0$ of their song is marked in the figure (approximately 140 Hz for subject T2, 164 Hz for both T1 and T3) and the overtone structure appears as horizontal bands. Varying degrees of vibrato can be observed, dependent upon the singer (*Figure 1*; see also longer spectrograms in *Appendix 1—figure 6* and *Appendix 1—figure 7*). Most of the energy in their song is concentrated in the overtones and no subharmonics (i.e., peaks at half-integer multiples of $f_0$) are observed. In contrast to these three singers, singer T4 performing in a non-Sygyt style exhibited a fundamental frequency of approximately 130 Hz, although significant energy additionally appears around 50–55 Hz, well below an expected subharmonic (*Appendix 1—figure 5*).

If we take a slice, that is a time-point from the spectrogram and plot the spectrum, we can observe the peaks to infer the formant structure from this representation of the sound (red-dashed lines in *Figure 1* and *Appendix 1—figure 4*). As the singers transition from normal singing to biphonation, we see that the formant structure changes significantly and the positions of formant peaks shift dramatically and rapidly. Note that considering time points before and after the transitions also provides an internal control for both normal and focused song types (*Appendix 1—figure 4*). Once in the biphonation mode, all three singers demonstrate overtones in a narrow spectral band around 1.5–2 kHz; we refer to this as the *focused state*. Specifically, *Figure 1* shows that not only is just a single or small group of overtones accentuated, but also that nearby ones are greatly attenuated: ±1 overtones are as much 15–35 dB and ±2 overtones are 35–65 dB below the central overtone. Whereas the energy in the low-frequency region associated with the first formant (below 500 Hz) is roughly constant between the normal-singing and focused states, there is a dramatic change in the spectrum for the higher formants above 500 Hz. In normal singing (i.e., prior to the focused state), spectral energy is distributed across several formants between 500 and 4000 Hz. In the focused state after the transition, the energy above 500 Hz becomes narrowly focused in the 1.5–2 kHz region, generating a whistle-like pitch that carries the song melody.

To assess the degree of focus objectively and quantitatively, we computed an energy ratio $e_R(f_L, f_H)$ that characterizes the relative degree of energy brought into a narrow band against the energy spread over the full spectrum occupied by human speech (see Materials and methods). In normal speech and singing, for $[f_L, f_H] = [1, 2\,\mathrm{kHz}]$, typically $e_R$ is small (i.e., energy is spread across the spectrum, not *focused* into that narrow region between 1 and 2 kHz). For the Tuvan singers, prior to a transition into a focused state, $e_R(1, 2)$ is similarly small. However once the transition occurs (red triangle in *Figure 1*), those values are large (upwards of 0.5 and higher) and sustained across time (*Appendix 1—figure 2* and *Appendix 1—figure 3*). For one of the singers (T2) the situation was more complex, as he created multiple focused formants (*Figure 1* middle panels and *Appendix 1—figure 6*, *Appendix 1—figure 8*). The second focused state was not explicitly dependent upon the first: The first focused state clearly moves and transitions between approximately 1.5–2 kHz (by 30%) while the second focused state remains constant at approximately 3–3.5 kHz (changing less than 1%). Thus the focused states are not harmonically related. Unlike the other singers, T2 not only has a second focused state, but also had more energy in the higher overtones (*Figure 1*). As such, singer T2 also exhibited a different $e_R$ time course, which took on values that could be relatively large even prior to the transition. This may be because he took multiple ways to approach the transition into a focused state (e.g., *Appendix 1—figure 9*).

Plotting spectra around the transition from normal to biphonation singing in a waterfall plot indicates that the sharp focused filter is achieved by merging two broader formants together ($F_2$ and $F_3$ in *Figure 2*; *Kob, 2004*). This transition into the focused state is fast (~40–60 ms), as are the shorter transitions within the focused state where the singer melodically changes the filter that forms the whistle-like component of their song (*Figure 1*, *Appendix 1—figure 8*).

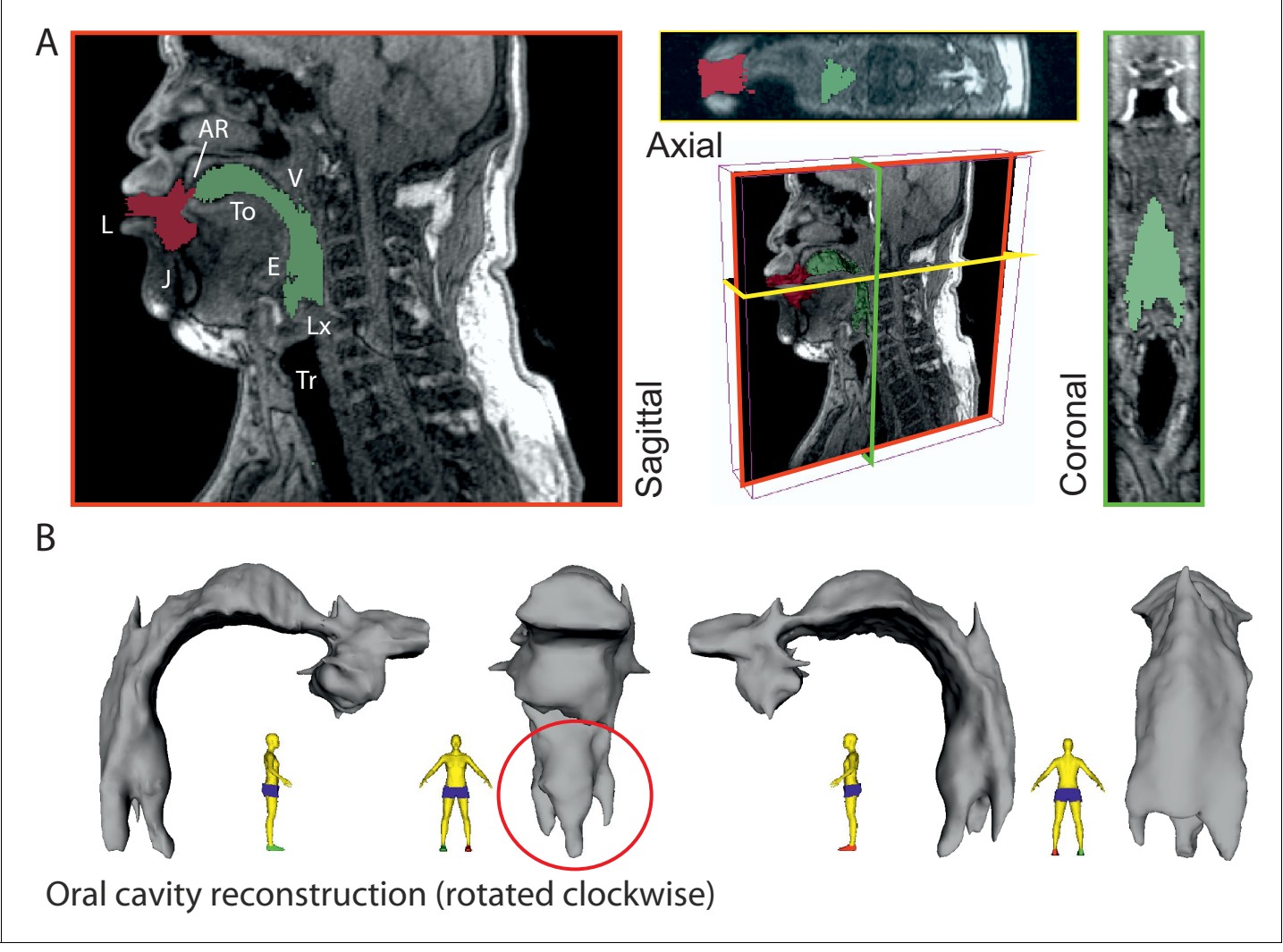

**Figure 3.** 3-D reconstruction of volumetric MRI data taken from singer T2 (Run3; see Appendix, including *Appendix 1—figure 18*). (**A**) Example of MRI data sliced through three different planes, including a pseudo-3D plot. Airspaces were determined manually (green areas behind tongue tip, red for beyond). Basic labels are included: L – lips, J – jaw, To– tongue, AR – alveolar ridge, V – velum, E – epiglottis, Lx – larynx, and Tr – trachea. The shadow from the dental post is visible in the axial view on the left hand side and stops near the midline leaving that view relatively unaffected. (**B**) Reconstructed airspace of the vocal tract from four different perspectives. The red circle highlights the presence of the piriform sinuses (*Dang and Honda, 1997*).

## Vocal tract MRI

While we can infer the shape of the formants in Khoomei by examining audio recordings, such analysis is not conclusive in explaining the mechanism used to achieve these formants. The working hypothesis was that vocal tract shape determines these formants. Therefore, it was crucial to examine the shape and dynamics of the vocal tract to determine whether the acoustic measurements are consistent with this hypothesis. To accomplish this, we obtained MRI data from one of the singers (T2) that are unique in two regards. First, there are two types of MRI data reported here: steady-state volumetric data *Figure 3* and *Appendix 1—figure 18*) and dynamic midsagittal images at several frames per second that capture changes in vocal tract position (*Figure 4A–B* and *Appendix 1—figure 20*). Second is that the dynamic data allow us to examine vocal tract changes as song transitions into a focused state (e.g., *Appendix 1—figure 20*).

The human vocal tract begins at the vocal folds and ends at the lips. Airflow produced by the vocal cords sets the air-column in the tract into vibration, and its acoustics determine the sound that emanates from the mouth. The vocal tract is effectively a tube-like cavity whose shape can be altered by several articulators: the jaw, lips, tongue, velum, epiglottis, larynx and trachea (*Figure 4C*).

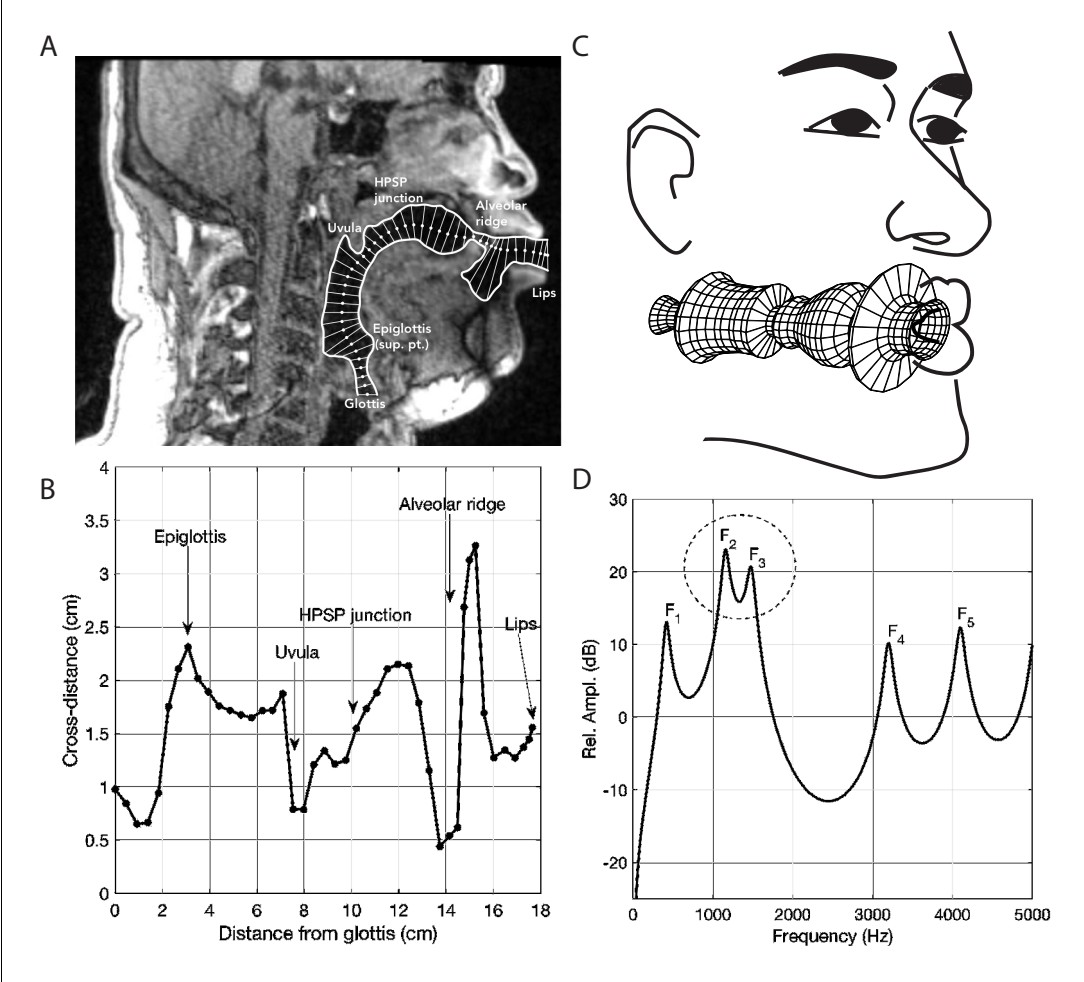

**Figure 4.** Analysis of vocal tract configuration during singing. (**A**) 2D measurement of tract shape. The inner and outer profiles were manually traced, whereas the centerline (white dots) was found with an iterative bisection technique. The distance from the inner to outer profile was measured along a line perpendicular to each point on the centerline (thin white lines). (**B**) Collection of cross-distance measurements plotted as a function of distance from the glottis. Area function can be computed directly from these values and is derived by assuming the cross-distances to be equivalent diameters of circular cross-sections (see Materials and methods). (**C**) Schematic indicating associated modeling assumptions, including vocal tract configuration as in panel B (adapted from ***Bunton et al. (2013)***, under a Creative Commons CC-BY license, https://creativecommons.org/licenses/by/4.0/). (**D**) Model frequency response calculated from the associated area function stemming from panels B and C. Each labeled peak can be considered a formant frequency and the dashed circle indicates merging of formants F2 and F3.

Producing speech or song requires that the shape of the vocal tract, and hence its acoustics, are precisely controlled (***Story, 2016***).

Several salient aspects of the vocal tract during the production of Khoomei can be observed in the volumetric MRI data. The most important feature however, is that there are two distinct and relevant constrictions when in the focused state, corresponding roughly to the uvula and alveolar ridge. Additionally, the vocal tract is expanded in the region just anterior to the alveolar ridge (***Figure 4A***). The retroflex position of the tongue tip and blade produces a constriction at 14 cm, and also results in the opening of this sublingual space. It is the degree of constriction at these two locations that is hypothesized to be the primary mechanism for creating and controlling the frequency at which the formant is *focused*.

## Modeling

Having established that the shape of vocal tract during Khoomei does indeed have two constrictions, consistent with observations from other groups, the primary goals of our modeling efforts were to

use the dynamic MRI data as morphological benchmarks and capture the merging of formants to create the focused states as well as the dynamic transitions into them. Our approach was to use a well-established linear "source/filter" model (e.g., *Stevens, 2000*) that includes known energy losses (*Sondhi and Schroeter, 1987*; *Story et al., 2000*; *Story, 2013*). Here, the vibrating vocals folds act as the broadband sound source (with the $f_0$ and associated overtone cascade), while resonances of the vocal tract, considered as a series of 1-D concatenated tubes of variable uniform radius, act as a primary filter. We begin with a first order assumption that the system behaves linearly, which allows us for a simple multiplicative relationship between the source and filter in the spectral domain (e. g., *Appendix 1—figure 10*).

Acoustic characteristics of the vocal tract can be captured by transforming the three-dimensional configuration (*Figure 3*) into a tube with variation in its cross-sectional area from the glottis to the lips (*Figure 4* and *Figure 5*). This representation of the vocal tract shape is called an *area function*, and allows for calculation of the corresponding frequency response function (from which the formant frequencies can be determined) with a one-dimensional wave propagation algorithm. Although the area function can be obtained directly from a 3D vocal tract reconstruction (e.g., *Story et al., 1996*), the 3D reconstructions of the Tuvan singer's vocal tract were affected by a large shadow from a dental post (e.g., see *Figure 4*) and were not amenable to detailed measurements of cross-sectional area. Instead, a cross-sectional area function was measured from the midsagittal slice of the 3D image set (see Materials and methods and Appendix for details). Thus, the MRI data provided crucial bounds for model parameters: the locations of primary constrictions and thereby the associated area functions.

The frequency response functions derived from the above static volumetric MRI data (e.g., *Figure 4D*) indicate that two formants $F_2$ and $F_3$ cluster together, thus enhancing both their amplitudes. Clearly, if $F_2$ and $F_3$ could be driven closer together in frequency, they would merge and form a single formant with unusually high amplitude. We hypothesize that this mechanism could be useful for effectively amplifying a specific overtone, such that it becomes a prominent acoustic feature in the sound produced by a singer, specifically the high frequency component of Khoomei.

Next, we used the model in conjunction with time-resolved MRI data to investigate how the degree of constriction and expansion at different locations along the vocal tract axis could be a mechanism for controlling the transition from normal to overtone singing and the pitch while in the focused state. These results are summarized in *Figure 5* (further details are in the Appendix). While the singers are in the normal song mode, there are no obvious strong constrictions in their vocal tracts (e.g., *Appendix 1—figure 11*). After they transition, in each MRI from the focused state, we observe a strong constriction near the alveolar ridge. We also observe a constriction near the uvula in the upper pharynx, but the degree of constriction here varies. If we examine the simultaneous audio recordings, we find that variations in this constriction are co-variant with the frequency of the focused formant. From this, we surmise that the mechanism for controlling the enhancement of voice harmonics is the degree of constriction near the alveolar ridge in the oral cavity (labeled $C_O$ in *Figure 5*), which affects the proximity of $F_2$ and $F_3$ to each other (*Appendix 1—figure 12*). Additionally, the degree of constriction near the uvula in the upper pharynx ($C_P$) controls the actual frequency at which $F_2$ and $F_3$ converge (*Appendix 1—figure 13*). Other parts of the vocal tract, specifically the expansion anterior to $C_O$, may also contribute since they also show small co-variations with the focused formant frequency (*Appendix 1—figure 14*). Further, a dynamic implementation of the model, as shown in *Appendix 1—figure 14*, reasonably captures the rapid transition into/out of the focused state as shown in *Figure 1*. Taken together, the model confirms and explains how these articulatory changes give rise to the observed acoustic effects.

To summarize, an overtone singer could potentially 'play' (i.e., select) various harmonics of the voice source by first generating a tight constriction in the oral cavity near the alveolar ridge, and then modulating the degree of constriction in the uvular region of the upper pharynx to vary the position of the focused formant, thereby generating a basis for melodic structure.

## Discussion

This study has shown that Tuvan singers performing Sygyt-style Khoomei exercise precise control of the vocal tract to effectively merge multiple formants together. They morph their vocal tracts so to create a sustained *focused* state that effectively filters an underlying stable array of overtones. This

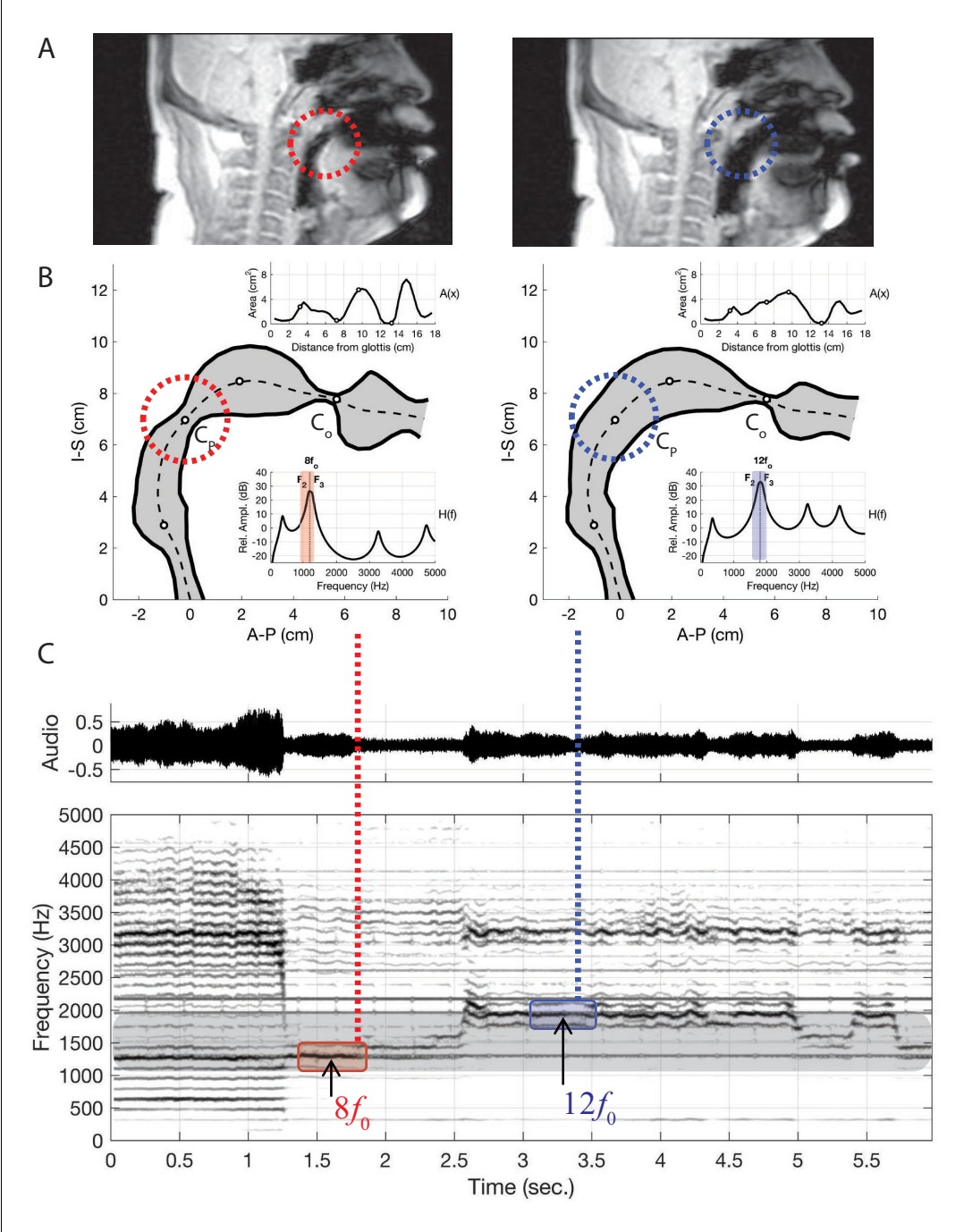

**Figure 5.** Results of changing vocal tract morphology in the model by perturbing the baseline area function $A_0(x)$ to demonstrate the merging of formants $F_2$ and $F_3$, atop two separate overtones as apparent in the two columns of panels A and B. (A) The frames from dynamic MRI with red and blue dashed circles highlighting the location of the key vocal tract constrictions. (B) Model-based vocal tract shapes stemming from the MRI data, including both the associated area functions (top inset) and frequency response functions (bottom inset). $C_O$ indicates the constriction near the alveolar ridge while $C_P$ the constriction near the uvula in the upper pharynx. (C) Waveform and corresponding spectrogram of audio from singer T2 (a spectrogram from the model is shown in *Appendix 1—figure 14*). Note that the merged formants lie atop either the 7th overtone (i.e., $8f_0$) or the 11th (i.e., $12f_0$).

focused filter greatly accentuates energy of a small subset of higher order overtones primarily in the octave-band spanning 1–2 kHz, as quantified by an energy ratio $e_R(1,2)$. Some singers are even capable of producing additional foci at higher frequencies. Below, we argue that a linear framework

(i.e., source/filter model, *Stevens, 2000*) appears sufficient to capture this behavior including the sudden transitions into a focused state, demonstrating that nonlinearities are not a priori essential. That is, since the filter characteristics are highly sensitive to vocal tract geometry, precise biomechanical motor control of the singers is sufficient to achieve a focused state without invoking nonlinearities or a second source as found in other vocalization types (e.g., *Herzel and Reuter, 1996*; *Fee et al., 1998*). Lastly, we describe several considerations associated with how focused overtone song produces such a salient percept by virtue of a pitch decoherence.

## Source or filter?

The notion of a focused state is mostly consistent with vocal tract filter-based explanations for biphonation in previous studies (e.g., *Bloothooft et al., 1992*; *Edgerton et al., 1999*; *Adachi and Yamada, 1999*; *Grawunder, 2009*), where terms such as an 'interaction of closely spaced formants', 'reinforced harmonics', and 'formant melting' were used. In addition, the merging of multiple formants is closely related to the 'singer's formant', which is proposed to arise around 3 kHz due to formants $F_3$–$F_5$ combining (*Story, 2016*), though this is typically broader and less prominent than the focused states exhibited by the Tuvans. Our results explain how this occurs and are also broadly consistent with *Adachi and Yamada (1999)* in that a constricted 'rear cavity' is crucial. However, we find that this rear constriction determines the pitch of the focused formant, whereas it is the 'front cavity' constriction near the alveolar ridge that produces the focusing effect (i.e., merging of formants $F_2$ and $F_3$).

Further, the present data appear in several ways inconsistent with conclusions from previous studies of Khoomei, especially those that center on effects that arise from changes in the source. Three salient examples are highlighted. First, we observed overtone structure to be highly stable, though some vibrato may be present. This contrasts the claim by *Levin and Edgerton (1999)* that "(t)o tune a harmonic, the vocalist adjusts the fundamental frequency of the buzzing sound produced by the vocal folds, so as to bring the harmonic into alignment with a formant'. That is, we see no evidence for the overtone 'ladder' being lowered or lifted as they suggested (note in *Figure 1*, $f_0$ is held nearly constant). Further, this stability argues against a transition into a different mode of glottal pulse generation, which could allow for a 'second source' (*Mergell and Herzel, 1997*). Second, a single sharply defined harmonic alone is not sufficient to get the salient perception of a focused state, as had been suggested by *Levin and Edgerton (1999)*. Consider *Appendix 1—figure 9*, especially at the 4 s mark, where the voicing is 'pressed'. *Pressed* phonation, also referred to as ventricular voice, occurs when glottal flow is affected by virtue of tightening the laryngeal muscles such that the ventricular folds are brought into vibration. This has the perceptual effect of adding a degree of roughness to the voice sound (*Lindestad et al., 2001*; *Edmondson and Esling, 2006*). There, a harmonic at 1.51 kHz dominates (i.e., the two flanking overtones are approximately 40 dB down), yet the song has not yet perceptibly transitioned. It is not until the cluster of overtones at 3–3.5 kHz is brought into focus that the perceptual effect becomes salient, perhaps because prior to the 5.3 s mark the broadband nature of those frequencies effectively masks the first focused state. Third, we do not observe subharmonics, which contrasts a prior claim (*Lindestad et al., 2001*) that "(t)his combined voice source produces a very dense spectrum of overtones suitable for overtone enhancement'. However, that study was focused on a different style of song called 'Kargyraa', which does not exhibit as clearly a focused state as in Sygyt.

## Linear versus nonlinear mechanisms

An underlying biophysical question is whether focused overtone song arises from inherently linear or nonlinear processes. Given that Khoomei consists of the voicing of two or more pitches at once and exhibits dramatic and fast transitions from normal singing to biphonation, nonlinear phenomena may seem like an obvious candidate (*Herzel and Reuter, 1996*). It should be noted that *Herzel and Reuter (1996)* go so far to define *biphonation* explicitly through the lens of nonlinearity. We relax such a definition and argue for a perceptual basis for delineating the boundaries of biphonation. Certain frog species exhibit biphonation, and it has been suggested that their vocalizations can arise from complex nonlinear oscillatory regimes of separate elastically coupled masses (*Suthers et al., 2006*). Further, the appearance of abrupt changes in physiological systems (as seen in *Figure 1*) has been

argued to be a flag for nonlinear mechanisms (*Goldberger et al., 2002*); for example, by virtue of progression through a bifurcation.

Our results present two lines of evidence that argue against Sygyt-style Khoomei arising primarily from a nonlinear process. First, the underlying harmonic structure of the vocal fold source appears highly stable through the transition into the focused state (*Figure 1*). There is little evidence of sub-harmonics. A source spectral structure that is comprised of an $f_0$ and integral harmonics would suggest a primarily linear source mechanism. Second is that our modeling efforts, which are chiefly linear in nature, reasonably account for the sudden and salient transition. That is, the model is readily sufficient to capture the characteristic that small changes in the vocal tract can produce large changes in the filter. Thereby, precise and fast motor control of the articulators in a linear framework accounts for the transitions into and out of the focused state. Thus, in essence, Sygyt-style Khoomei could be considered a linear means to achieve biphonation. Connecting back to nonlinear phonation mechanisms in non-mammals, our results provide further context for how human song production and perception may be similar and/or different relative to that of non-humans (e.g., *Doolittle et al., 2014*; *Kingsley et al., 2018*).

Nevertheless, features that appear transiently in spectrograms do provide hints of source nonlinearity, such as the brief appearance of subharmonics in some instances (*Appendix 1—figure 15B*). This provides an opportunity to address the limitations of the current modeling efforts and to highlight future considerations. We suggest that further analysis (e.g., *Theiler et al., 1992*; *Tokuda et al., 2002*; *Kantz and Schreiber, 2004*) of Khoomei audio recordings may help to inform the model and might better capture focused filter sharpness and the origin of secondary focused states. Several potential areas for improvement are: nonlinear source–filter coupling (*Titze et al., 2008*); a detailed model of glottal dynamics (e.g., ratio of open/closed phases in glottal flow [*Grawunder, 2009*; *Li and Hou, 2017*], and periodic vibrations in $f_0$); inclusion of piriform sinuses as side-branch resonators (*Dang and Honda, 1997*; *Titze and Story, 1997*); inclusion of the 3-D geometry; and detailed study of the front cavity (e.g., lip movements) that may be used by the singer to maintain control of the focused state and to make subtle manipulations.

## Perceptual consequences of overtone focusing

Although this study did not directly assess the percept associated with these vocal productions, the results raise pressing questions about how the spectro-temporal signatures of biphonic Khoomei described here create the classical perception of Sygyt-style Khoomei as two distinct sounds (*Aksenov, 1973*). The first, the low-pitched drone, which is present during both the normal singing and the focused-state biphonation intervals, reflects the pitch associated with $f_0$, extracted from the harmonic representation of the stimulus. It is well established that the perceived pitch of a broadband sound comprised of harmonics reflects the $f_0$ derived primarily from the perceptually resolved harmonics up to about $10f_0$ (*Bernstein and Oxenham, 2003*). The frequency resolution of the peripheral auditory system is such that these low-order harmonics are individually resolved by the cochlea, and it appears that such filtering is an important prerequisite for pitch extraction associated with that common $f_0$. The second sound, the high-pitched melody, is present only during the focused-state intervals and probably reflects a pitch associated with the focused formant. An open question, however, is why this focused formant would be perceived incoherently as a separate pitch (*Shamma et al., 2011*), when it contains harmonics at multiples of $f_0$. The auditory system tends to group together concurrent harmonics into a single perceived object with a common pitch (*Roberts et al., 2015*), and the multiple formants of a sung or unsung voice are not generally perceived as separate sounds from the low harmonics.

The fact that the focused formant is so narrow apparently leads the auditory system to interpret this sound as if it were a separate tone, independent of the low harmonics associated with the drone percept, thereby effectively leading to a pitch decoherence. This perceptual separation could be attributable to a combination of both bottom-up (i.e., cochlear) and top-down (i.e., perceptual) factors. From the bottom-up standpoint, even if the focused formant is broad enough to encompass several harmonic components, the fact that it consists of harmonics at or above $10 f_0$ (i.e., the 1500 Hz formant frequency represents the 10th harmonic of a 150 Hz $f_0$) means that these harmonics will not be spectrally resolved by cochlear filtering (*Bernstein and Oxenham, 2003*). Instead, the formant will be represented as a single spectral peak, similar to the representation of a single pure tone at the formant frequency. Although the interaction of harmonic components at this cochlear

location will generate amplitude modulation at a rate equal to the $f_0$ (*Plack and Oxenham, 2005*), it has been argued that a common $f_0$ is a weak cue for binding low- and high-frequency formants (*Culling and Darwin, 1993*). Rather, other top-down mechanisms of auditory-object formation may play a more important role in generating a perception of two separate objects in Khoomei. For example, the rapid onsets of the focused formant may enhance its perceptual separation from the constant drone (*Darwin, 1984*). Further, the fact that the focused formant has a variable frequency (i.e., frequency modulation, or FM) while the drone maintains a constant $f_0$ is another difference that could facilitate their perceptual separation. Although it has been argued that FM differences between harmonic sounds generally have little influence on their perceived separation (*Darwin, 2005*), others have reported enhanced separation in the special case in which one complex was static and the other had applied FM (*Summerfield and Culling, 1992*) – similar to the first and second formants during the Tuvan focused state.

The perceptual separation of the two sounds in the Tuvan song might be further affected by a priori expectations about the spectral qualities of vocal formants (*Billig et al., 2013*). Because a narrow formant occurs so rarely in natural singing and speech, the auditory system might be pre-disposed against perceiving it as a phonetic element, limiting its perceptual integration with the other existing formants. Research into 'sine-wave speech' provides some insights into this phenomenon. When three or four individual frequency-modulated sinusoids are presented at formant frequencies in lieu of natural formants, listeners can, with sufficient training, perceive the combination as speech (*Remez et al., 1981*). Nevertheless, listeners largely perceive these unnatural individual pure tones as separate auditory objects (*Remez et al., 2001*), much like the focused formant in Khoomei. Further research exploring these considerations would help close the production–perception circle underlying the unique percept arising from Tuvan throat song.

## Materials and methods

### Acoustical recordings

Recordings were made at York University (Toronto, ON, Canada) in a double-walled acoustic isolation booth (IAC) using a Zoom H5 24-bit digital recorder and an Audio-Technica P48 condenser microphone. A sample rate of 96 kHz was used. Spectral analysis was done using custom-coded software in Matlab. Spectrograms were typically computed using 4096 point window segments with 95% fractional overlap and a Hamming window. Harmonics (black circles in *Figure 1*) were estimated using a custom-coded peak-picking algorithm. Estimated formant trends (red dashed lines in *Figure 1*) were determined using a lowpass interpolating filter built into Matlab's digital signal processing toolbox with a scaling factor of 10. From this trend, the peak-picking was reapplied to determine 'formant' frequencies (red 'x's in *Figure 1*). This process could be repeated across the spectrogram to track overtone and formant frequency/strength effectively, as shown in *Appendix 1—figure 1*.

To quantify the focused states, we developed a dimension-less measure $e_R(f_L, f_H)$ to represent the energy ratio of that spanning a frequency range $f_H - f_L$ relative to the entire spectral output. This can be readily computed from the spectrogram data as follows. First take a 'slice' from the spectrogram and convert spectral magnitude to linear ordinate and square it (as intensity is proportional to pressure squared). Then integrate across frequency, first for a limited range spanning $[f_L, f_H]$ (e.g., 1–2 kHz) and then for a broader range of $[0, f_{max}]$ (e.g., 0–8 kHz; 8 kHz is a suitable maximum as there is little acoustic energy in vocal output above this frequency). The ratio of these two is then defined as $e_R$, and takes on values between 0 and 1. This can be expressed more explicitly as:

$$e_R(f_L, f_H) = \left( \frac{\int_{f_L}^{f_H} P(f)\, df}{\int_0^{f_{max}} P(f)\, df} \right)^2 \tag{1}$$

where $P$ is the magnitude of the scaled sound pressure, $f$ is frequency, and $f_L$ and $f_H$ are filter limits for considering the focused state. The choice of $[f_L, f_H] = [1, 2]$ kHz has the virtue of spanning an octave, which also closely approximates the 'seventh octave' from about C6 to C7. $e_R$ did not depend significantly upon the length of the fast Fourier transform (FFT) window. Values of $e_R$ for the waveforms used in *Figure 1* are shown in *Appendix 1—figures 2* and *3*.

## MRI acquisition and volumetric analysis

MRI images were acquired at the York MRI Facility on a 3.0 Tesla MRI scanner (Siemens Magnetom TIM Trio, Erlangen, Germany), using a 12-channel head coil and a neck array. Data were collected with the approval of the York University Institutional Review Board. The participant was fitted with an MRI compatible noise-cancelling microphone (Optoacoustics, Mazor, Israel) mounted directly above the lips. The latency of the microphone and noise-cancelling algorithm was 24 ms. Auditory recordings were made in QuickTime on an iMac during the scans to verify performance.

Images were acquired using one of two paradigms, static or dynamic. Static images were acquired using a T1-weighted 3D gradient echo sequence in the sagittal orientation with 44 slices centered on the vocal tract, TR = 2.35 ms, TE = 0.97 ms, flip angle = 8 degrees, FoV = 300 mm, and a voxel dimension of 1.2 × 1.2×1.2 mm. Total acquisition time was 11 s. The participant was instructed to begin singing a tone, and to hold it in a steady state for the duration of the scan. The scan was started immediately after the participant began to sing and had reached a steady state. Audio recordings verified a consistent tone for the duration of the scan. Dynamic images were acquired using a 2D gradient echo sequence. A single 10.0 mm thick slice was positioned in a sagittal orientation along the midline of the vocal tract, TR = 4.6 ms, TE = 2.04 ms, flip angle = 8 degrees, FoV = 250 mm, and a voxel dimension of 2.0 × 2.0×10.0 mm. One hundred measurements were taken for a scan duration of 27.75 s. The effective frame rate of the dynamic images was 3.6 Hz. Audio recordings were started just prior to scanning. Only subject T2 participated in the MRI recordings. The participant was instructed to sing a melody for the duration of the scan, and took breaths as needed.

For segmentation (*Figure 3*), 3D MRI images (Run1; see Appendix) were loaded into Slicer (version 4.6.2 r25516). The air-space in the oral cavity was manually segmented using the segmentation module, identified and painted in slice by slice. Careful attention was paid to the parts of the oral cavity that were affected by the artifact from the dental implant. The air cavity was manually repainted to be approximately symmetric in this region using the coronal and axial view (*Figure 3A*). Once completely segmented, the sections were converted into a 3D model and exported as a STL file. This mesh file was imported into MeshLab (v1.3.4Beta) for cleaning and repairing the mesh. The surface of the STL was converted to be continuous by removing non-manifold faces and then smoothed using depth and Laplacian filters. The mesh was then imported into Meshmixer where further artifacts were removed. This surface-smoothed STL file was finally reimported into Slicer, generating the display in *Figure 3B*.

## Computational modeling

Measurement of the cross-distance function is illustrated in *Figure 4*. The inner and outer profiles of the vocal tract were first determined by manual tracing of the midsagittal image. A 2D iterative bisection algorithm (*Story, 2007*) was then used to find the centerline within the profiles extending from the glottis to the lips, as shown by the white dots in *Figure 4A*. Perpendicular to each point on the centerline, the distance from the inner to outer profiles was measured to generate the cross-distance function shown in *Figure 4B*; the corresponding locations of the anatomic landmarks shown in the midsagittal image are also indicated on the cross-distance function.

The cross-distance function, $D(x)$, can be transformed to an approximate area function, $A(x)$, with the relation $A(x) = kD^{\alpha}(x)$, where $k$ and $\alpha$ are a scaling factor and exponent, respectively. If the elements of $D(x)$ are considered to be diameters of a circular cross-section, $k = (\pi/4)$ and $\alpha = 2$. Although other values of $k$ and $\alpha$ have been proposed to account for the complex shape of the vocal tract cross-section (*Heinz and Stevens, 1964*; *Lindblom and Sundberg, 1971*; *Mermelstein, 1973*), there is no agreement on a fixed set of numbers for each parameter. Hence, the circular approximation was used in this study to generate an estimate of the area function. In *Figure 4C*, the area function is plotted as its tubular equivalent, where the radii $D(x)/2$ were rotated about an axis to generate circular sections from the glottis to the lips.

The associated frequency response of that area function is shown in *Figure 4D* and was calculated with a transmission line approach (*Sondhi and Schroeter, 1987*; *Story et al., 2000*), which included energy losses due to yielding walls, viscosity, heat conduction, and acoustic radiation at the lips. Side branches such the piriform sinuses were not considered in detail in this study. The first five formant frequencies (resonances), $F_1, \ldots, F_5$, were determined by finding the peaks in the frequency

response functions with a peak-picking algorithm (*Titze et al., 1987*) and are located at 400, 1065, 1314, 3286, and 4029 Hz, respectively.

To examine changes in pitch, a particular vocal tract configuration was manually 'designed (*Appendix 1—figure 6*) such that it included constrictive and expansive regions at locations similar to those measured from the singer (i.e., *Figure 4*), but to a less extreme degree. We henceforth denote this area function as $A_0(x)$, and it generates a frequency response with widely spaced formant frequencies ($F_{1...5} = [529, 1544, 2438, 3094, 4236]$ Hz), essentially a neutral vowel. In many of the audio signals recorded from the singer, the fundamental frequency, $f_o$ (i.e., the vibratory frequency of the vocal folds), was typically about 150 Hz. The singer then appeared to enhance one of the harmonics in the approximate range of $8f_o \ldots 12f_o$. Taking the 12th harmonic ($12 \times 150 = 1800$ Hz) as an example target frequency (dashed line in the frequency response shown in *Figure 5c*), the area function $A_0(x)$ was iteratively perturbed by the acoustic-sensitivity algorithm described in *Story (2006)* until $F_2$ and $F_3$ converged on 1800 Hz and became a single formant peak in the frequency response. Additional details on the perturbation process leading into *Figure 5* are detailed in the Appendix.

## Acknowledgements

A heartfelt thank you to Huun Huur Tu, without whom this study would not have been possible. Input/suggestions from Ralf Schlueter, Greg Huber, Dorothea Kolossa, Chris Rozell, Tuomas Virtanen, and the reviewers are gratefully acknowledged. Support from York University, the Fields Institute for Research in Mathematical Sciences, and the Kavli Institute of Theoretical Physics is also gratefully acknowledged. CB was supported by the Natural Sciences and Engineering Research Council of Canada (NSERC) Grant RGPIN-430761–2013. The identification of specific products or scientific instrumentation does not constitute endorsement or implied endorsement on the part of the author, Department of Defense, or any component agency. The views expressed in this article are those of the authors and do not reflect the official policy of the Department of Army/Navy/Air Force, the Department of Defense, or the U.S. Government.

## Additional information

### Funding

| Funder | Grant reference number | Author |
| --- | --- | --- |
| Natural Sciences and Engineering Research Council of Canada | RGPIN-430761-2013 | Christopher Bergevin |

The funders had no role in study design, data collection and interpretation, or the decision to submit the work for publication.

### Author contributions

Christopher Bergevin, Conceptualization, Data curation, Software, Formal analysis, Investigation, Visualization, Methodology; Chandan Narayan, Conceptualization, Investigation; Joy Williams, Jennifer KE Steeves, Investigation, Methodology; Natasha Mhatre, Investigation, Visualization, Methodology; Joshua GW Bernstein, Investigation, Writing - original draft; Brad Story, Software, Formal analysis, Investigation, Visualization, Methodology

### Author ORCIDs

Christopher Bergevin https://orcid.org/0000-0002-4529-399X
Natasha Mhatre http://orcid.org/0000-0002-3618-306X
Jennifer KE Steeves https://orcid.org/0000-0002-7487-4646
Brad Story https://orcid.org/0000-0002-6530-8781

### Ethics

Human subjects: Data were collected with approval of the York University Institutional Review Board (IRB protocol to Prof Jennifer Steeves) This study was approved by the Human Participants Review

Board of the Office of Research Ethics at York University (certificate #2017-132) and adhered to the tenets of the Declaration of Helsinki. All participants gave informed written consent and consent to publish prior to their inclusion in the study.

## Decision letter and Author response
Decision letter https://doi.org/10.7554/eLife.50476.sa1
Author response https://doi.org/10.7554/eLife.50476.sa2

# Additional files

## Supplementary files
• Transparent reporting form

## Data availability
All data files (audio and imaging), as well as the relevant analysis software, are available via https://doi.org/10.5061/dryad.cvdncjt14.

The following dataset was generated:

| Author(s) | Year | Dataset title | Dataset URL | Database and Identifier |
|---|---|---|---|---|
| Bergevin C | 2020 | Overtone focusing in biphonic Tuvan throat singing | https://doi.org/10.5061/dryad.cvdncjt14 | Dryad Digital Repository, 10.5061/dryad.cvdncjt14 |

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

## Appendix 1

This appendix contains supporting information for the document *Overtone focusing in biphonic Tuvan throat singing* by Bergevin et al. Citations here refer to the bibliography of the main document. First (***Methodological considerations***), we include several methodological components associated with the quantitative analysis of the waveforms, helping illustrate different approaches towards characterizing the acoustic data and rationale underlying control measures. Second (***Additional waveform analyses***), we include additional plots to support results and discussion in the main text. For example, different spectrograms are presented, as are analyses for additional waveforms. This section also helps to provide additional context for a second independent focused state. The third section (***Additional modeling analysis figures***) details theoretical components leading into the results of the computational model and how the MRI data constrain the key parameters, justifying arguments surrounding the notion of formant merging. Fourth (***Instability in focused state***), some speculative discussion and basic modeling aspects are presented with regard to the notion of instabilities present in the motor control of the focused state. In the fifth section (***Additional MRI analysis figures***), images stemming from the MRI data are presented. Last, the final three sections detail accessing the acoustic waveforms, MRI data files, and waveform analysis (Matlab-based) software via an online repository.

## Methodological considerations

### Overtone and formant tracking

To facilitate quantification of the waveforms, we custom-coded a peak-picking/tracking algorithm to analyze the time-frequency representations produced by the spectrogram. *Appendix 1—figure 1* shows an example of the tracking of the overtones (red dots) and formants (grayscale dots; intensity coded by relative magnitude as indicated by the colorbar). This representation provides an alternative view (compared to *Figure 1*) to help demonstrate that, by and large, the overtone structure is highly consistent throughout, while the formant structure varies significantly across the transition.

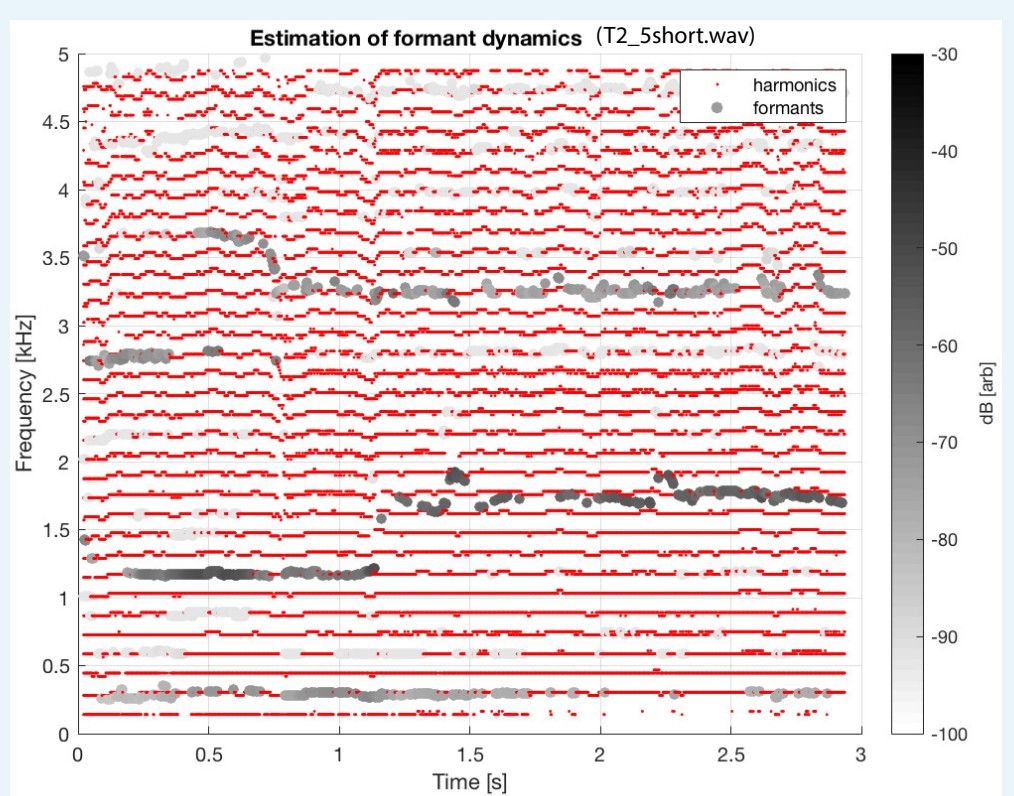

**Appendix 1—figure 1.** Same as *Figure 1* (middle left panel; subject T2, same sound file as shown in the middle panel of *Figure 1*), except with overtones and estimated formant structure tracked across time.

## Quantifying focused states

*Appendix 1—figures 2* and *3* show calculation of the energy ratio $e_R$ used as a means to quantify the degree of focus. For *Appendix 1—figure 2*, the waveforms are the same as those shown in *Figure 1* (those with slightly different axis limits). In general, we found that $e_R(1, 2)$ provided a clear means to distinguish the focused state, as values were close to zero prior to the transition and larger/sustained beyond the transition. Singer T2 was an exception. *Appendix 1—figure 3* is for singer T2, using the same file (i.e., the transition point into the focused state at between 6 and 7 s in this figure is the same as that shown in the middle panel of *Figure 1*), but with an expanded timescale to illustrate the larger $e_R$ values prior to the transition. This is due to the relatively large amount of energy present between 2.5–4 kHz. We also explored $e_R(1, 2)$ values in a wide range of phonetic signals, such as child and adult vocalizations, other singing styles (e.g., opera), non-Tuvan singers (e.g., Westerners) performing 'overtone singing', and older recordings of Tuvan singers. In general, it was observed that $e_R(1, 2)$ was relatively large and sustained across time for focused overtone song, whereas the value was close to zero and/or highly transient for other vocalizations.

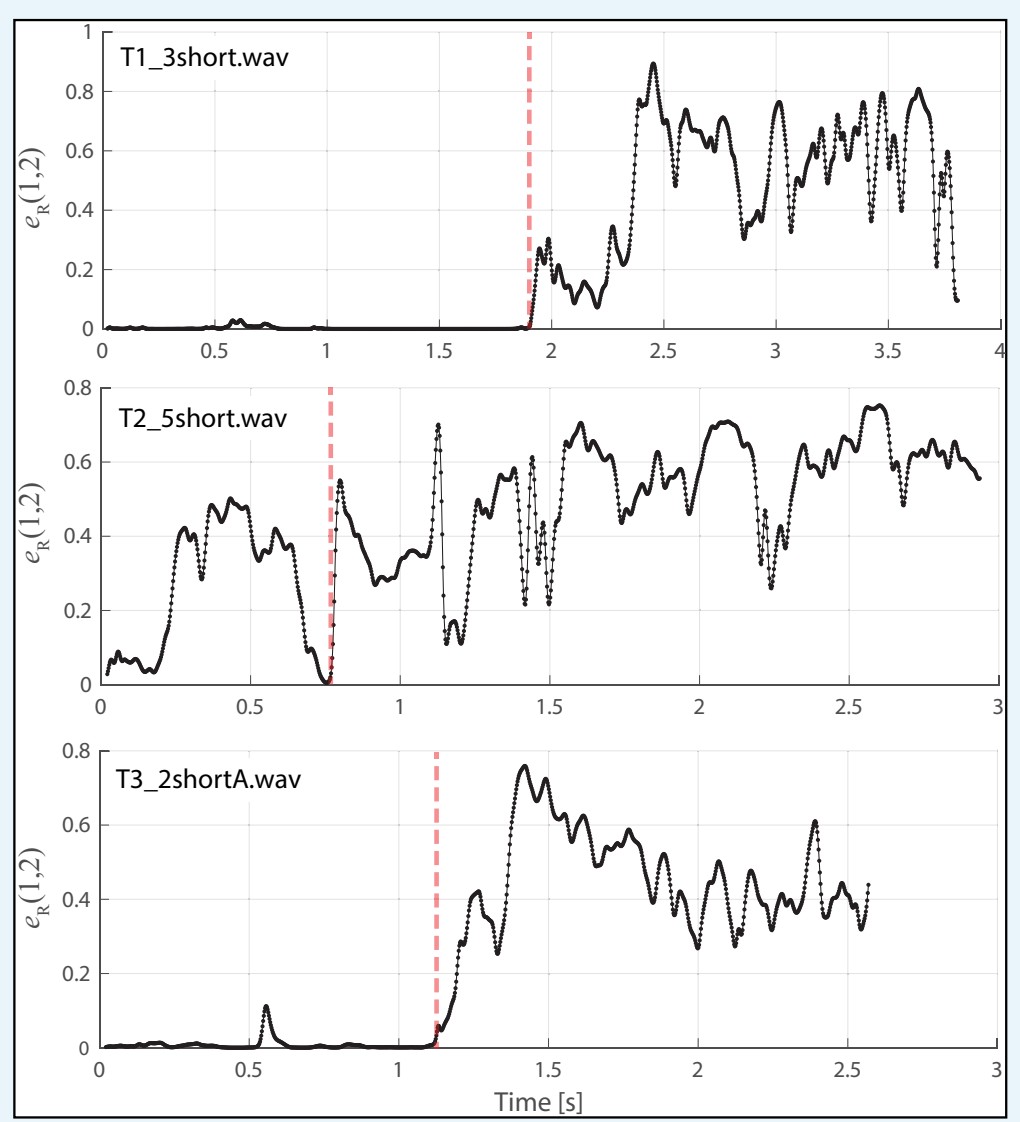

**Appendix 1—figure 2.** Same data/layout as in *Figure 1* but now showing $e_R(1,2)$ as defined in the 'Materials and methods'. These plots show the energy ratio focused between 1–2 kHz. Vertical red dashed lines indicate approximate time of transition into the focused state. An expanded timescale is also shown for singer T2 (middle panel) in *Appendix 1—figure 3*.

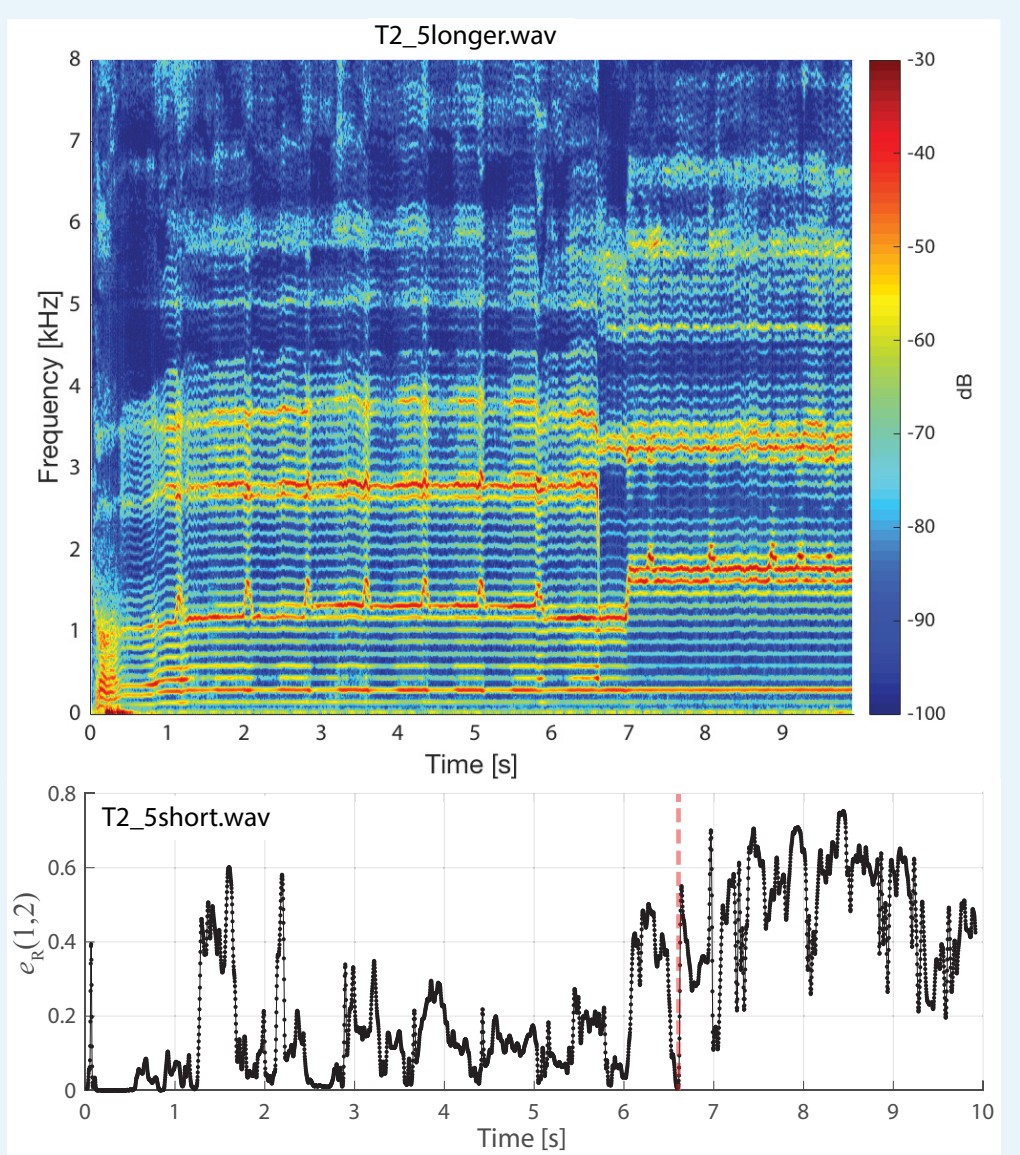

**Appendix 1—figure 3.** Similar to *Figure 2* for singer T2 (middle panel), except an expanded time scale is shown to demonstrate the earlier dynamics as this singer approaches the focused state (see T2_5longer.wav).

## Control measurements

The waveforms from the Tuvan singers provide an intrinsic degree of *control* (i.e., voicing not in the focused state). Similar to *Figure 1*, *Appendix 1—figure 4* shows the spectra prior to the transition into the focused state. Although relatively narrow harmonics can be observed, they tend to occur below 1 kHz. Such is consistent with our calculations of $e_R(1,2)$: prior to a transition into a focused state, this value is close to zero. The exception is singer T2, who instead shows a relatively large amount of energy about 1.8–3 kHz that may have some sort of masking effect (see 'Discussion' in the main text,and the 'Pressed transition' section below). In addition, Tuvan singer T4, who used a non-Sygyt style (*Appendix 1—figure 5*) , can also effectively be considered a 'control'.

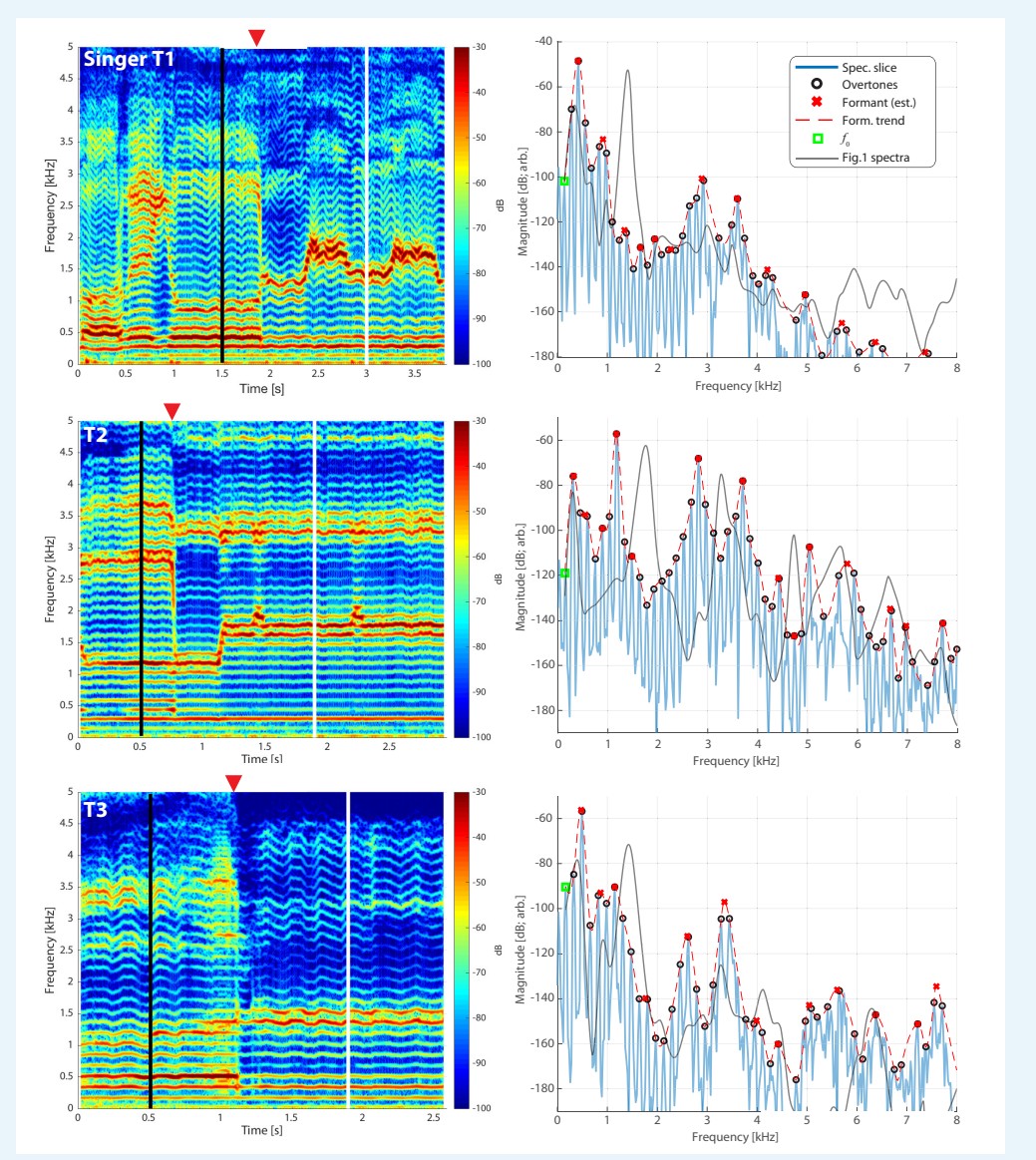

**Appendix 1—figure 4.** Stemming directly from *Figure 1*, the right-hand column now shows a spectrum from a time point prior to transition into the focused state (as denoted by the vertical black lines in the left column). The shape of the spectra from *Figure 1* is also included for reference.

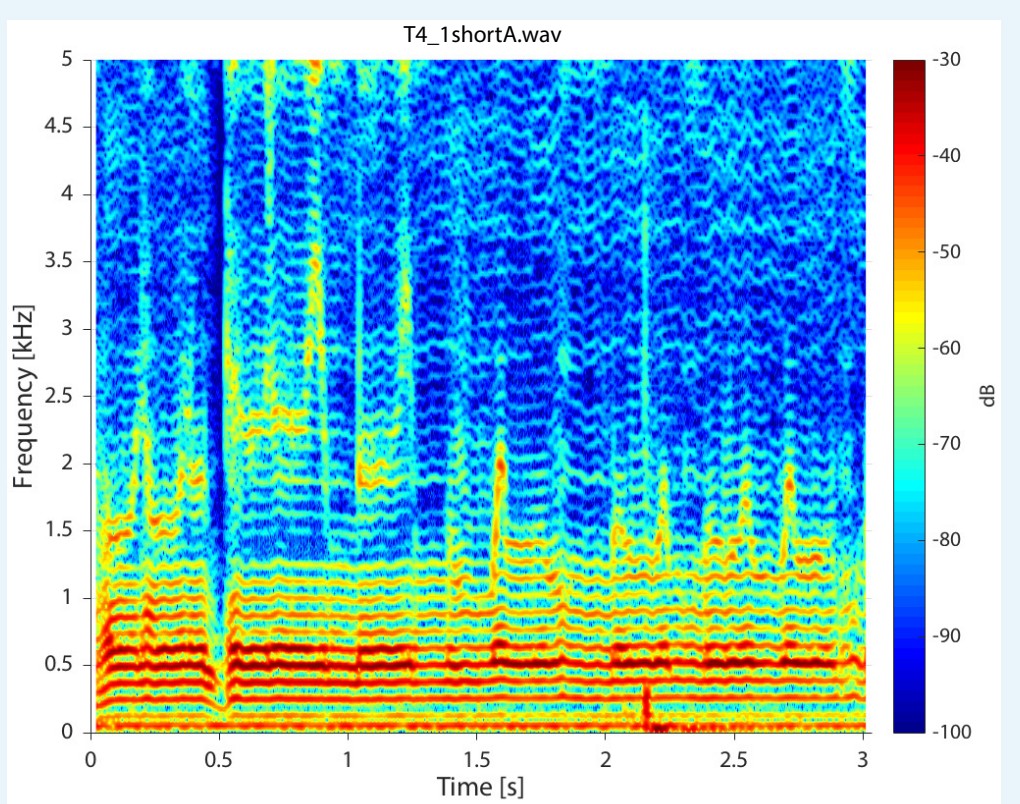

**Appendix 1—figure 5.** Spectrogram for singer T4 singing in non-Sygyt style (first song segment of T2_4shortA.wav sound file). For the spectrogram, 4096 point windows were used for the fast Fourier transform (FFT) with 95% fractional overlap and a Hamming window.

## Additional waveform analyses

### Other spectrograms

*Appendix 1—figure 5* shows spectrogram from singer T4 (T4_shortA.wav) singing in non-Sygyt style. While producing a distinctive sound, note the relative lack of energy above approximately 1 kHz. *Appendix 1—figure 6* shows a spectrogram from singer T2 (T2_5.wav) over a longer timescale than that shown in *Figure 1*. Similarly for *Appendix 1—figure 7*, but for singer T1. Both of these plots provide a spectral-temporal view of how the singer maintains and modulates the song over the course of a single exhalation. Note both the sudden transitions into different maintained pitches and the briefer transient excursions.

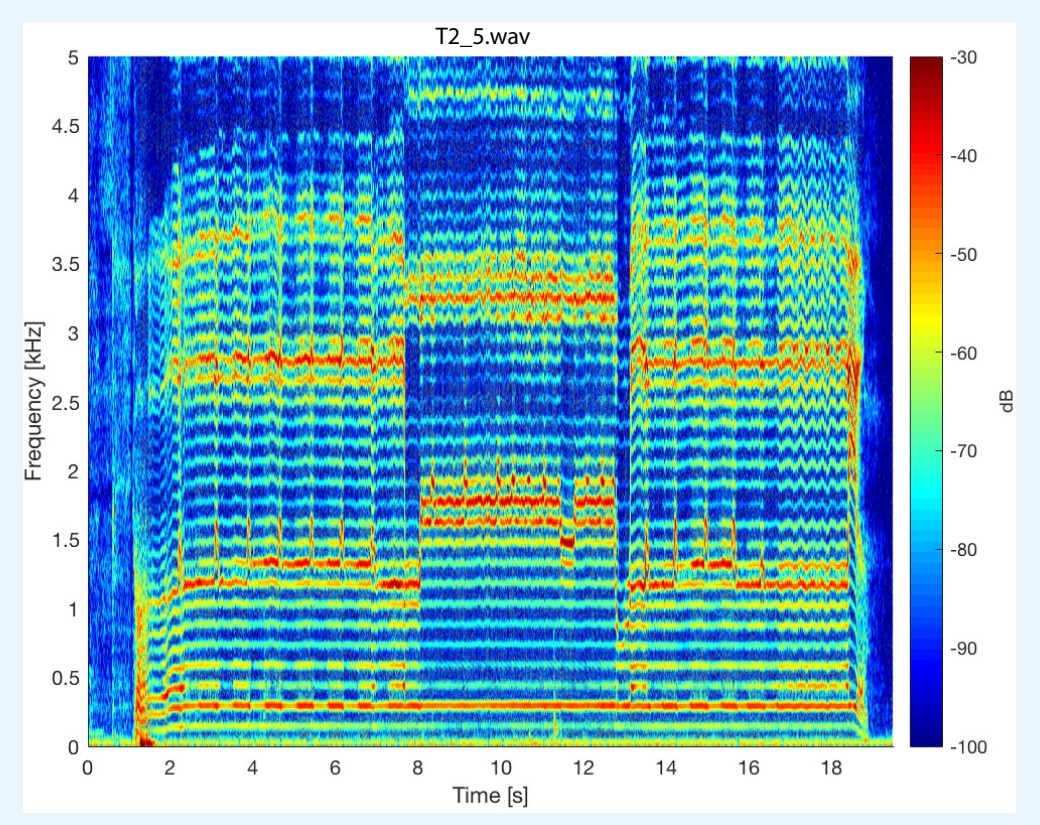

**Appendix 1—figure 6.** Spectrogram of the entire T2_5.wav sound file. The sample rate was 96 kHz. The analysis parameters used were the same as those used for *Figure 5*.

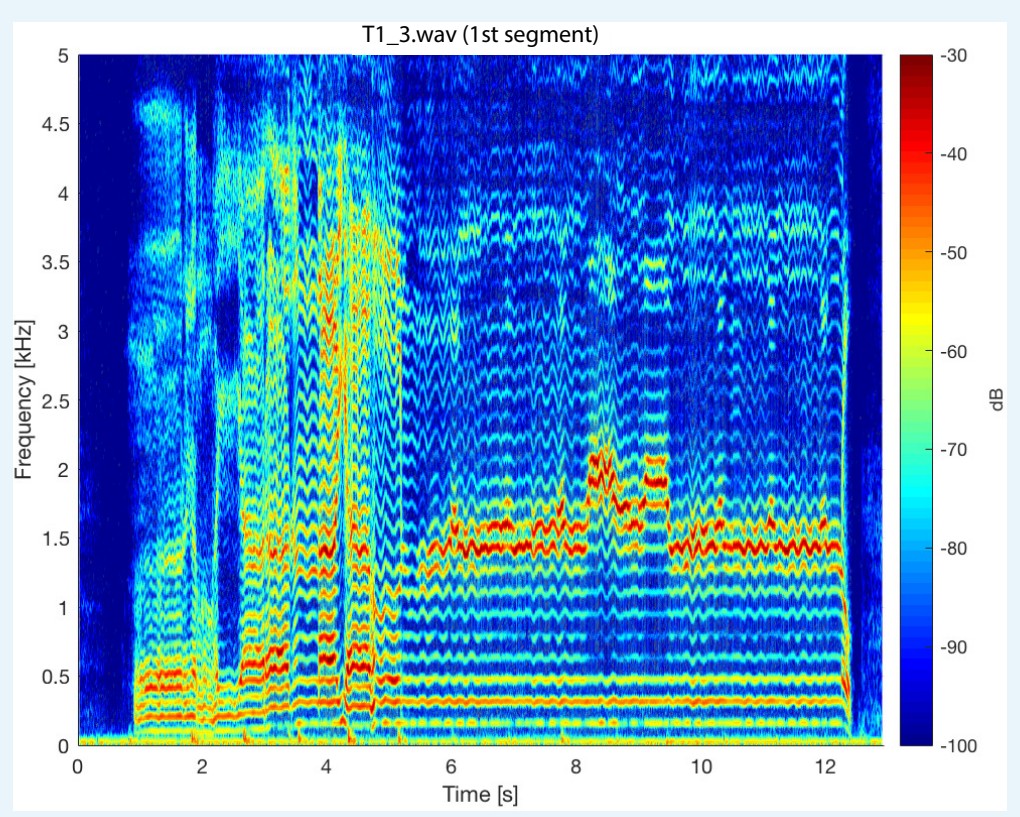

**Appendix 1—figure 7.** Spectrogram of the first song segment of the T1_3.wav sound file. The analysis parameters used were the same as those for *Figure 5*.

## Second independent focused state

*Appendix 1—figure 8* shows another example of a transition in Sygyt-style song for singer T2, clearly showing a second focused state about 3–3.5 kHz. Two aspects merit highlighting. First, the spectral peaks are not harmonically related: at $t = 4.5$ s, the first focused state is at 1.36 kHz and the other at 3.17 kHz (far from to 2.72 kHz as expected). Second, during the singer-induced pitch change at 3.85 s, the two peaks do not move in unison. Although not ruling out correlations between the two focused states, these observations suggest that they are not simply nor strongly related to one another.

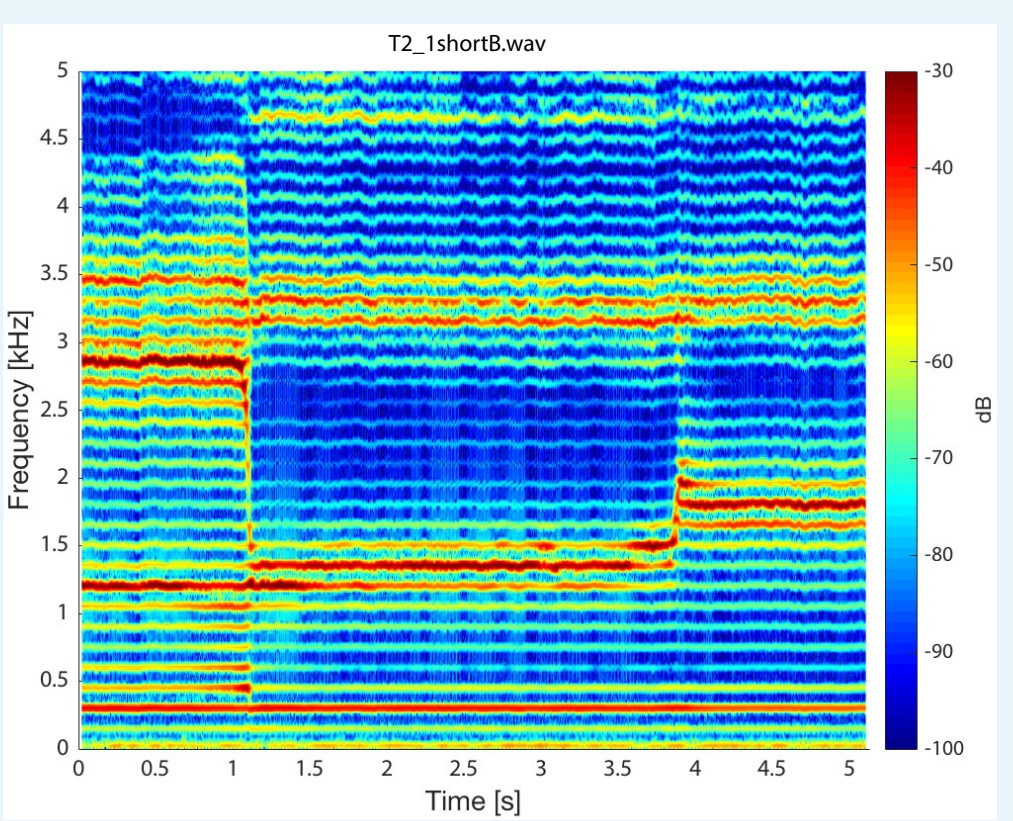

**Appendix 1—figure 8.** Singer T2's transition into a focused state. Note that while the first focused state transitions from approximately 1.36 to 1.78 kHz, the second state remains nearly constant, decreasing only slightly from 3.32 to 3.17 kHz (T2_1shortB.wav).

## Pressed transition

*Appendix 1—figure 9* shows a spectrogram and several spectral slices for the sound file in which the voicing was 'pressed' (*Adachi and Yamada, 1999*; *Edmondson and Esling, 2006*) prior to the transition into the focused state. That is, prior to the 1.8 s mark, voicing is relatively normal. But after that point (prior to the transition into the focused state around 5.4 s, substantial energy appears between 2–4 kHz along with a degree of vibrato. Note, however, that there is no change to the overall overtone structure (e.g., no emergence of subharmonics). The spectrum at $t = 4.0\text{s}$, prior to the transition, provides a useful comparison back to *Levin and Süzükei (2006)*. Specifically, one particular overtone is singled out and highly focused, yet the broadband cluster of overtones about 2.5–4 kHz effectively mask it. It is not until about the 5.4 s mark, when those higher overtones are also brought into focus, that a salient perception of the Sygyt-style emerges.

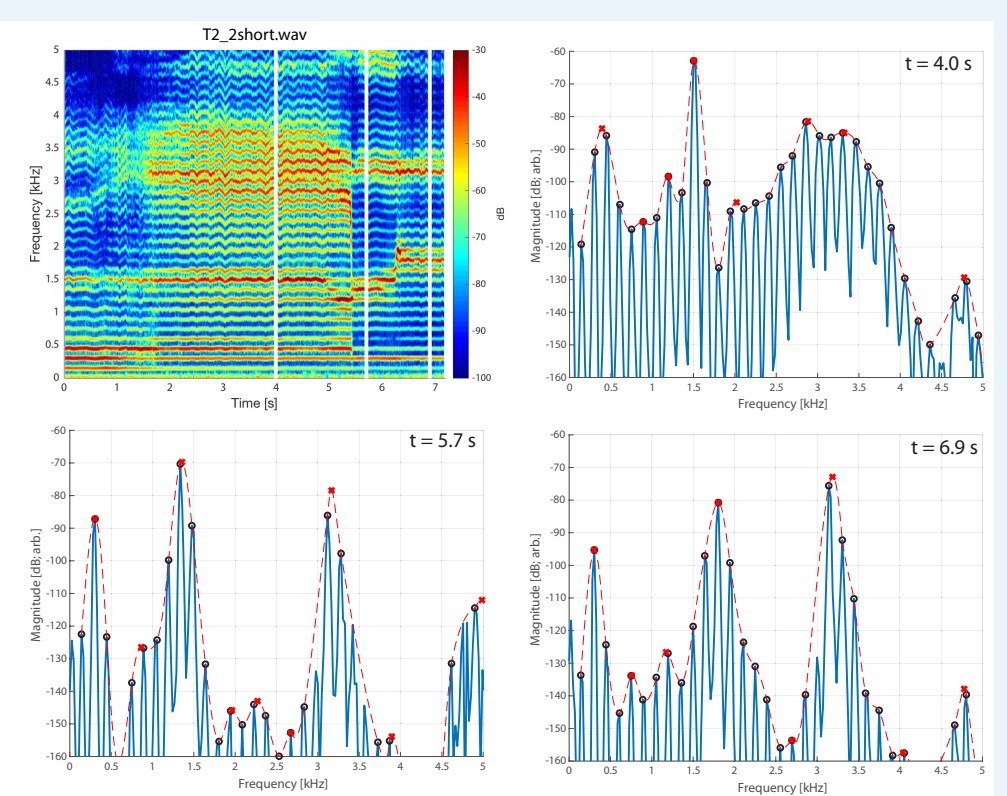

**Appendix 1—figure 9.** Spectrogram of singer T2 exhibiting pressed voicing heading into transition to focused state (T2_2short.wav).

## Additional modeling analysis figures

The measurement of the cross-distance function (as described in the 'Materials and methods'), along with calculation of the frequency response from an estimate of the area function, suggested that constrictions of the vocal tract in the region of the uvula and alveolar ridge may play a significant role in controlling the spectral focus generated by the convergence of $F_2$ and $F_3$. Assuming that an overtone singer configures the vocal tract to merge these two formants deliberately such that, together, they enhance the amplitude of a selected harmonic of the voice source, the aim was to investigate how the vocal tract can be systematically shaped with precisely placed constrictions and expansions to both merge $F_2$ and $F_3$ into a focused cluster and move the cluster along the frequency axis to allow for selection of a range of voice harmonics.

*Appendix 1—figure 11b* shows the same area function as that in *Appendix 1—figure 11a* (see 'Materials and methods') but plotted by extending the equivalent radius of each cross-sectional area, outward and inward, along a line perpendicular to the centerline measured from the singer (see *Figure 4A*), resulting in an inner and outer outline of the vocal tract shape as indicated by the thick black lines. The measured centerline is also shown in the figure, along with anatomic landmarks. As this does not represent a true midsagittal plane, it will be referred to here as a *pseudo-midsagittal* plot (*Story et al., 2001*).

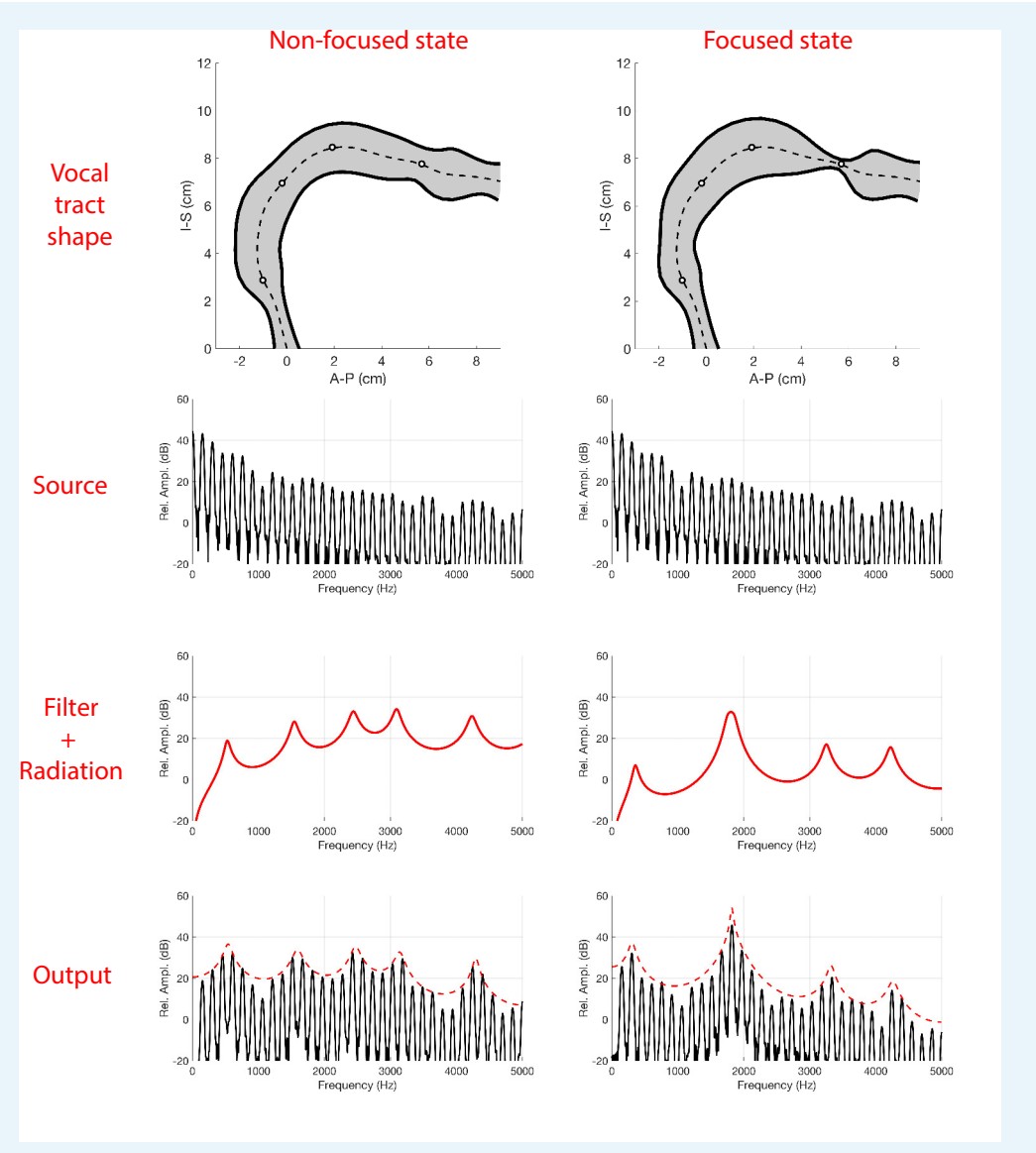

**Appendix 1—figure 10.** Overview of source/filter theory, as advanced by *Stevens (2000)*. The left column shows normal phonation, whereas the right indicates one example of a focused state.

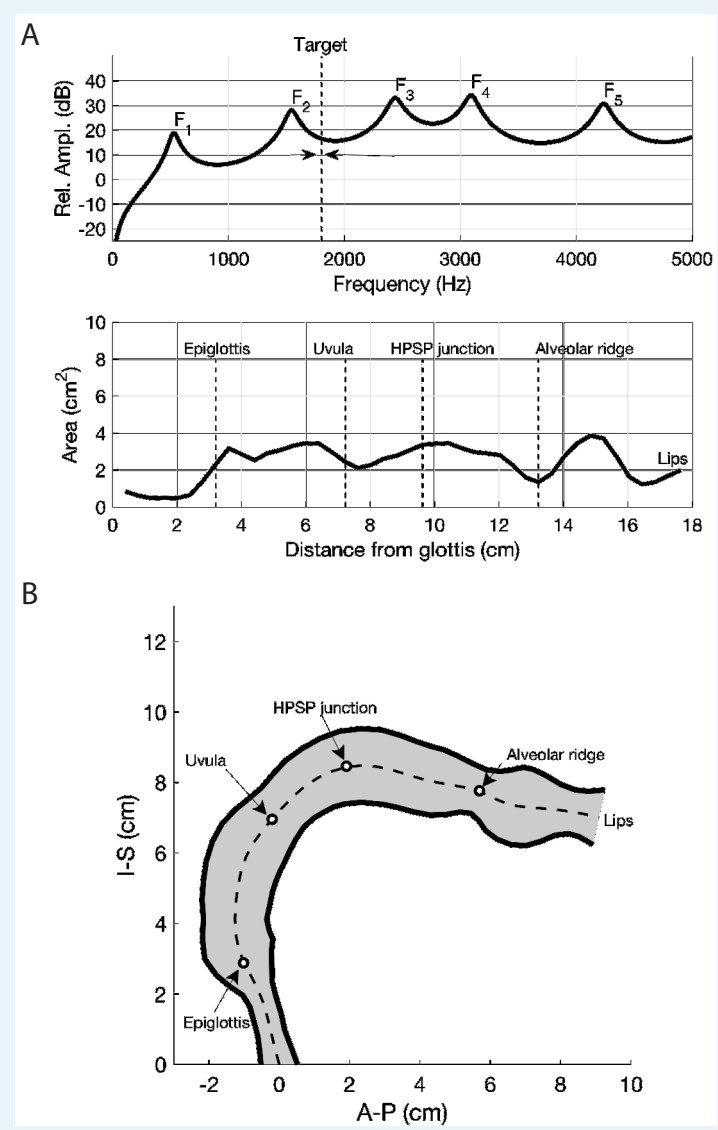

**Appendix 1—figure 11.** Setup of the baseline vocal tract configuration used in the modeling study. (**a**) The area function ($A_0(x)$) is in the lower panel and its frequency response is in the upper panel. (**b**) The area function from (**a**) is shown as a pseudo-midsagittal plot (see text).

*Appendix 1—figure 12a* shows the new area function and frequency response generated by the perturbation process, whereas the pseudo-midsagittal plot is shown in *Appendix 1—figure 12b*. Relative to the shape of $A_0(x)$ (shown as the thin gray line), the primary modification is a severe constriction imposed between 12.5–13.5 cm from the glottis, essentially at the alveolar ridge. Although the line thickness might suggest that the vocal tract is occluded in this region, the minimum cross-sectional area is 0.09 cm². There is also a more moderate constriction at about 5 cm from the glottis, and a slight expansion between 7–10.5 cm from the glottis. The frequency response in upper panel of *Appendix 1—figure 12a* demonstrates that the new area function was successful in driving $F_2$ and $F_3$ together to form a single formant peak centered at 1800 Hz, which is at least 15 dB higher in amplitude than any of the other formants. Exactly the same process was used to generate area functions for which $F_2$ and $F_3$ converge on the target harmonic frequencies: $8f_o, 9f_o, 10f_o, 11f_o = 1200, 1350, 1500, 1650$ Hz, respectively. The results, along with those from the previous figure for $12f_o$, are shown in *Appendix 1—figure 13*. The collection of frequency responses in the upper panel of *Appendix 1—figure 13b* shows that $F_2$ and $F_3$ successfully converged to become one formant peak in each of the cases, and their locations on the frequency axis are

centered around the specified target frequencies. The corresponding area functions in the lower panel suggests that the constriction between 12.5–13.5 cm from the glottis (alveolar ridge region) is present in roughly the same form for all five cases. By contrast, an increasingly severe constriction must be imposed in the region between 6–8.5 cm from the glottis (uvular region) in order to shift the target frequency (i.e., the frequency at which $F_2$ and $F_3$ converge) downward through progression of specified harmonics. Coincident with this constriction is a progressively larger expansion between 14–15.5 cm from the glottis, which probably assists in positioning the focal regions of $F_2$ and $F_3$ downward. It can also be noted that the area function that generates a focus at $8f_o$ (1200 Hz; thinnest line) is most similar to the one generated from the cross-distance measurements (i.e., **Figure 4c**). In both, there are constrictions located at about 7.5 cm and 13 cm from the glottis; the expansions in the lower pharynx and oral cavity are also quite similar. The main difference is the greater expansion of the region between 8–13 cm from the glottis in the acoustically derived area function.

On the basis of the results, a mechanism for controlling the enhancement of voice harmonics can be proposed: the degree of constriction near the alveolar ridge in the oral cavity (labeled $C_o$ in **Figure 5** of the main text) controls the proximity of $F_2$ and $F_3$ to each other, whereas the degree of constriction near the uvula in the upper pharynx, $C_p$, controls the frequency at which $F_2$ and $F_3$ converge (the expansion anterior to $C_o$ may also contribute). Thus, an overtone singer could potentially 'play' (i.e., select) various harmonics of the voice source by first generating a tight constriction in the oral cavity near the alveolar ridge to generate the focus of $F_2$ and $F_3$, and then modulating the degree of constriction in the uvular region of the upper pharynx to position the focus on a selected harmonic.

This proposed mechanism of controlling the spectral focus is supported by observation of vocal tract changes based on dynamic MRI data sets. Using this approach, midsagittal movies of the Tuvan singer were acquired in which each image represented approximately 275 ms. Shown in **Figure 5** is a comparison of vocal tract configurations derived with the acoustic-sensitivity algorithm (middle panels) to image frames from an MRI-based movie (upper panels) associated with the points in time indicated by the vertical lines superimposed across the waveform and spectrogram in the lower part of the figure. The image frames were chosen such that they appeared to be representative of the singer placing the spectral focus at $8f_o$ (left) and $12f_o$ (right), respectively, based on the evidence available in the spectrogram. The model-based vocal tract shape in the upper left panel, derived for a spectral focus of $8f_o$ (1200 Hz), exhibits a fairly severe constriction in the uvular region, similar to the constrictive effect that can be seen in the corresponding image frame (middle left). Likewise, the vocal tract shape derived for a spectral focus of $12f_o$ (1800 Hz) (upper right) and the image frame just below it both demonstrate an absence of a uvular constriction. Thus, the model-based approach generated vocal tract shapes that appear to possess characteristics similar to those produced by the singer, and provides support for the proposed mechanism of spectral focus control.

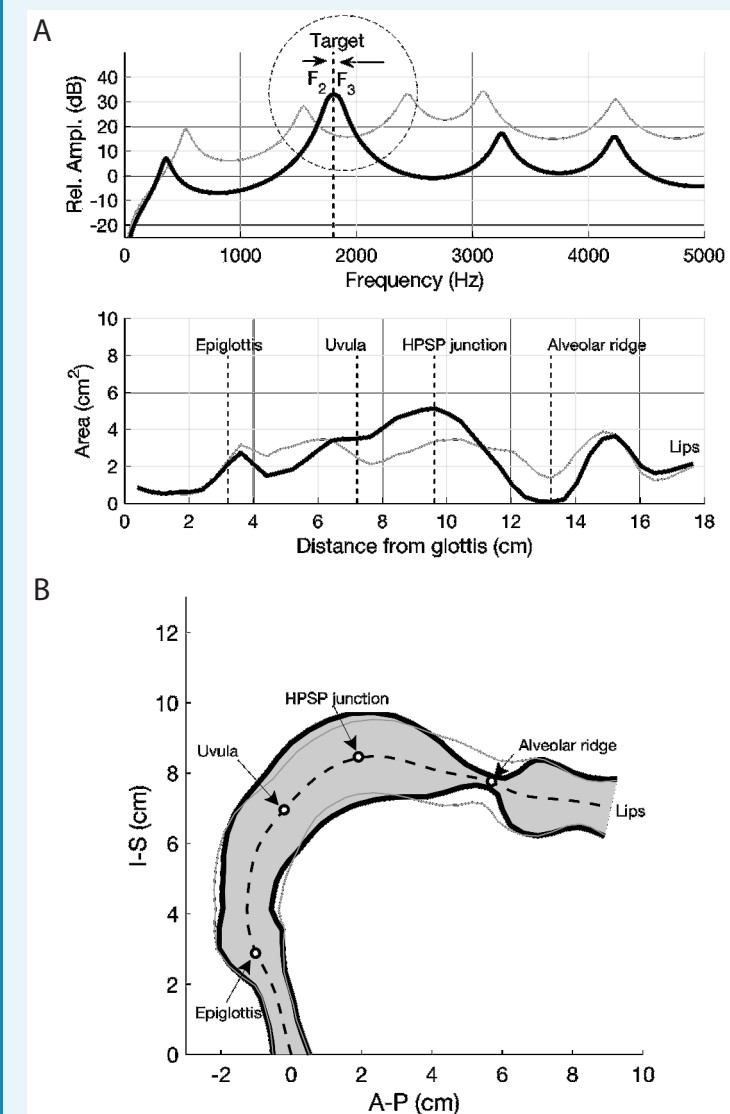

**Appendix 1—figure 12.** Results of perturbing the baseline area function $A_0(x)$ so that $F_2$ and $F_3$ converge on 1800 Hz. (**a**) Perturbed area function (thick black line) and the corresponding frequency response; for comparison, the baseline area function is also shown (thin gray line). The frequency response shows the convergence of $F_2$ and $F_3$ into one high amplitude peak centered around 1800 Hz. (**b**) Pseudo-midsagittal plot of the perturbed area function (thick black line) and the baseline area function (thin gray line).

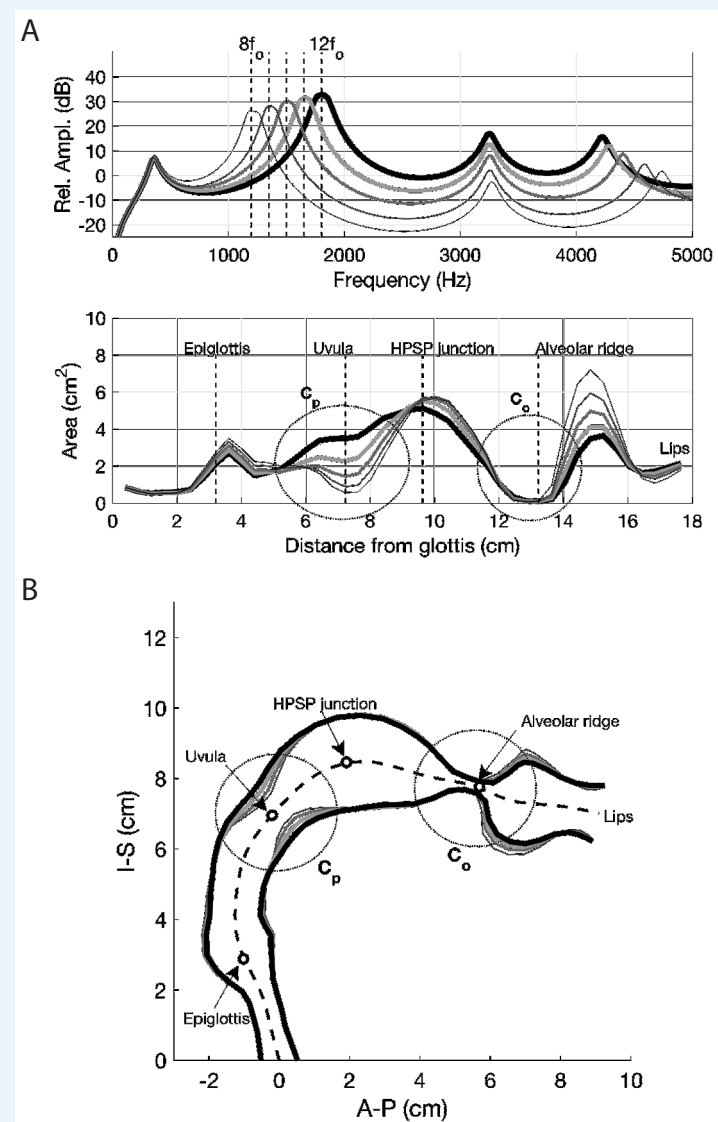

**Appendix 1—figure 13.** Results of perturbing the baseline area function $A_0(x)$ so that $F_2$ and $F_3$ converge on 1200, 1350, 1500, 1650, and 1800 Hz. (**A**) Perturbed area functions and corresponding frequency responses; line thicknesses and gray scale are matched in the upper and lower panels. (**B**) Pseudo-midsagittal plot of the perturbed area functions. The circled regions (dotted) denote constrictions that control the proximity of $F_2$ and $F_3$ to each other and the frequency at which they converge.

## Second focused state

Given that singer T2 was the subject for the MRI scans and uniquely exhibited a second focused state (e.g., **Appendix 1—figure 8**), the model was also utilized to explore how multiple states could be achieved. Two possibilities appear to be the sharpening of formant F4 alone, or the merging of F4 and F5 (**Appendix 1—figure 14**). However, it is unclear how reasonable those vocal tract configurations may be and further study is required.

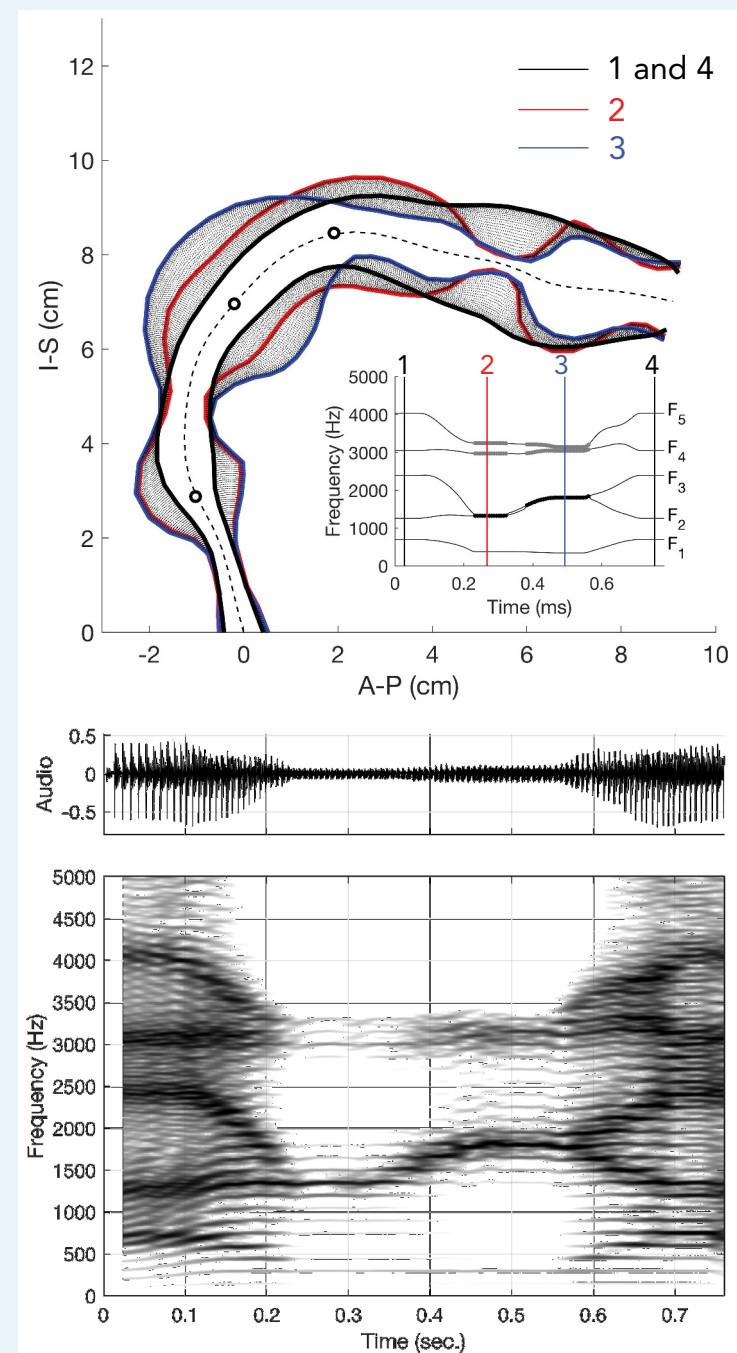

**Appendix 1—figure 14.** Similar to *Figure 5*, but additional manipulations were considered to create a second focused state by merging F4 and F5, as exhibited by singer T2 (see middle row in *Figure 1*). In addition, the spectrogram shown here is from the model (not the singer's audio). See also *Appendix 1—figure 20* for connections back to dynamic MRI data.

## Animations and synthesized song

Animations and audio clips demonstrating various quantitative aspects of the model are included in the data files posted to datadryad.org. Specifically they are:

o Animation (no sound) of vocal tract changes during transition into focused state and subsequent pitch changes – `Medley 0 to5 1 cluster. mp4Audioclipof simulatedsong`

```
o Audio clip of simulated song - Medley 0 to5 1 cluster s im.wav
```

### Instability in focused state

*Appendix 1—figure 15* and *Appendix 1—figure 16* show that brief transient instabilities in the focused state can and do regularly occur. Specifically, it can be observed that there are brief transient lapses while the singer is maintaining the focused overtone condition, thereby providing insight into how focus is actively maintained. One possible explanation is control by virtue of biomechanical feedback, where the focused state can effectively be considered to be an unstable equilibrium point, akin to balancing a ruler vertically on the palm of your hand. An alternative consideration might be that singers learn to create additional quasi-stable equilibrium points (e.g., *Appendix 1—figure 17*). The sudden transitions observed (*Figure 1*) could then be likened to two-person cheerleading moves such as a 'cupie', where one person standing on the ground suddenly throws another up vertically and has them balancing atop their shoulders or upward-stretched hands. A simple proposed model for the transition into the focused state is shown in *Appendix 1—figure 17*. There, a stable configuration of the vocal tract would be the low point (pink ball). Learning to achieve a focused state would give rise to additional stable equilibria (red ball), which may be more difficult to maintain. Considerations along these lines, combined with a model for biomechanical control (e.g., *Sanguineti et al., 1998*), can lead to testable predictions specific to when a highly experienced singer is maintaining balance about the transition point into/out of a focused state (e.g., T2_4.wav audio file).

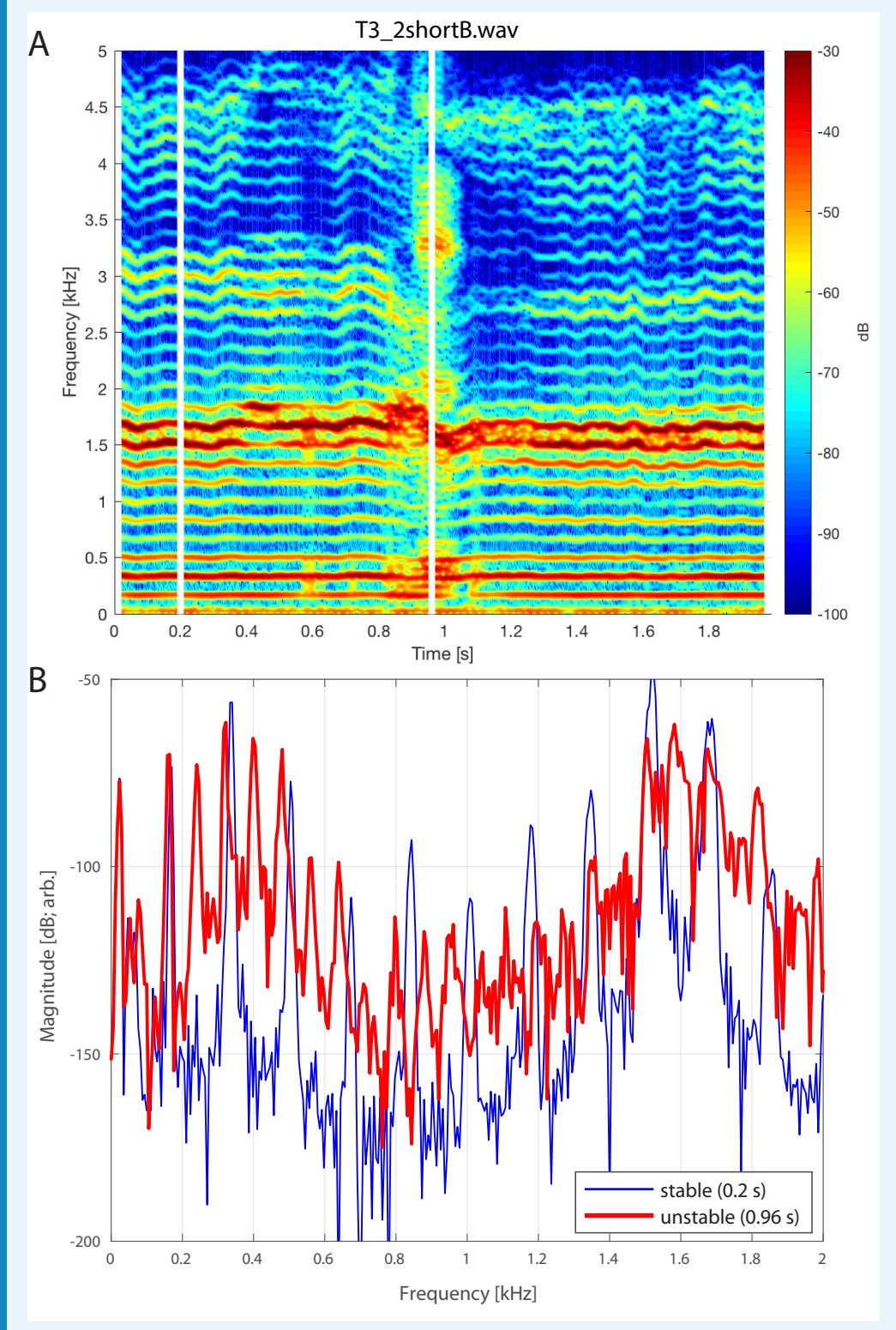

**Appendix 1—figure 15.** Brief instability in the focused state. (**A**) Spectrogram of singer T3 during period during which the focused state briefly falters (T3_2shortB.wav, extracted from around the 33 s mark of T3_2.wav). (**B**) Spectral slices taken at two different time points (vertical white lines in panel A at 0.2 and 0.96 s), the latter falling in the transient unstable state. Note that while there is little change in $f_0$ between the two periods (170 Hz versus 164

Hz), the unstable period shows a period doubling such that the subharmonic (i.e., $f_0/2$) and associated overtones are now present, indicative of nonlinear phonation.

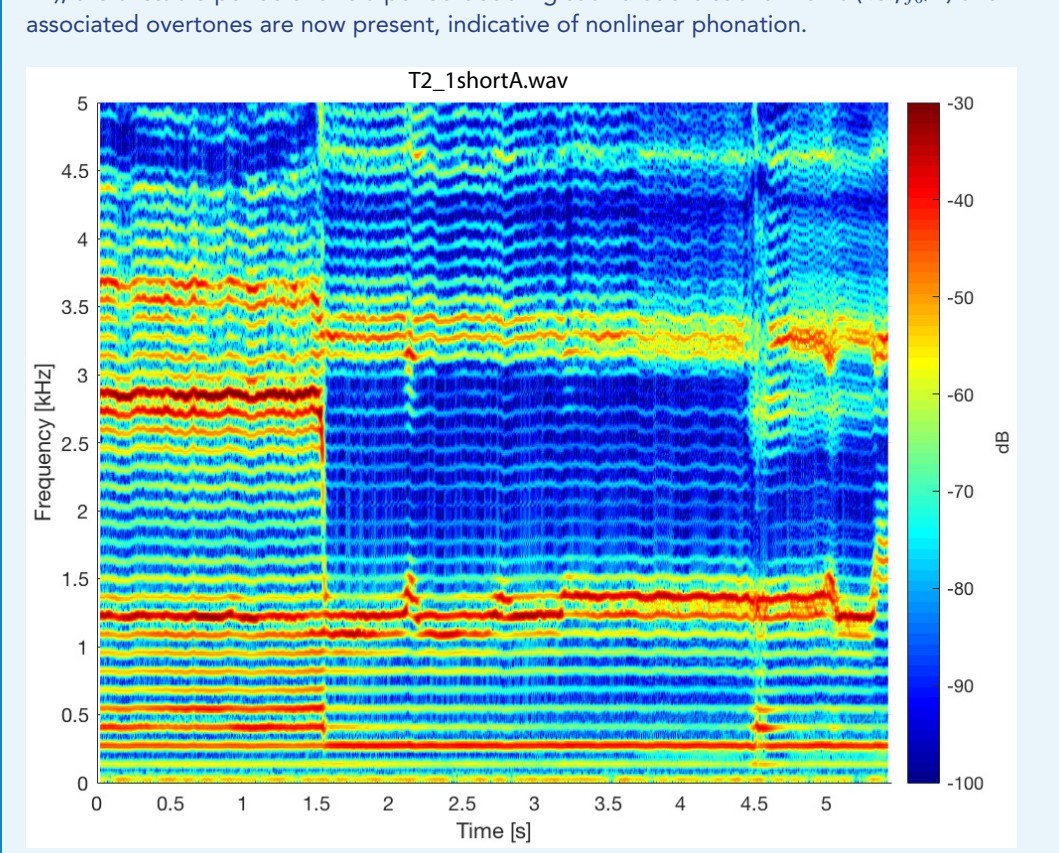

**Appendix 1—figure 16.** Spectrogram of singer T2 (T2_1shortA.wav) about a transition into a focused state. Note that there is a slight instability around 4.5 s.

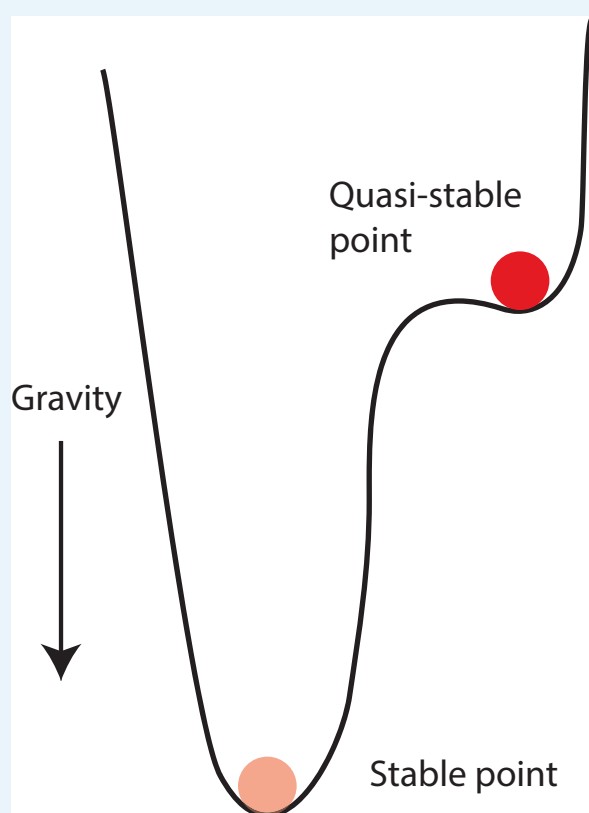

**Appendix 1—figure 17.** Schematic illustrating a simple possible mechanical analogy (ball confined to a potential well) for the transition into a focused state.

## Additional MRI analysis figures

### Volumetric data

An example of the volumetric data (arranged as tiled midsagittal slices) is shown in *Appendix 1—figure 18*. Note that the NMR artifact resulting from the presence of a dental post is apparently lateralized to one side.

   *Appendix 1—figure 19* shows a spectrogram of audio segment (extracted from Run3Vsound.wav) associated with the volumetric scan shown in *Appendix 1—figure 18*. Segments both with and without the scanner noise are shown.

### Vocal tract shape and associated spectrograms

Examples of the vocal tract taken during the dynamic MRI runs (i.e., midsagittal only) are shown for very different representative time points in *Appendix 1—figure 20*.

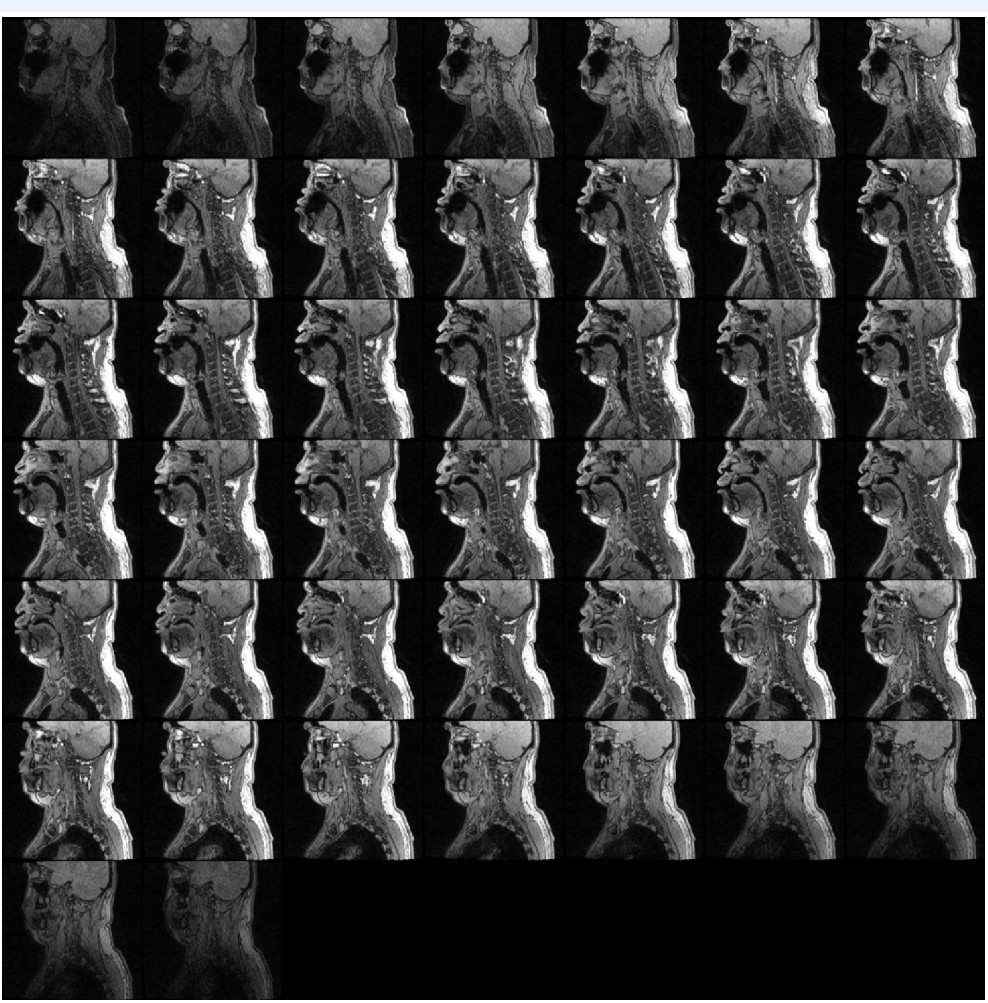

**Appendix 1—figure 18.** Mosaic of single slices from the volumetric MRI scan (Run3) of subject T2 during focused overtone state. Spectrogram of corresponding audio shown in *Appendix 1—figure 19*.

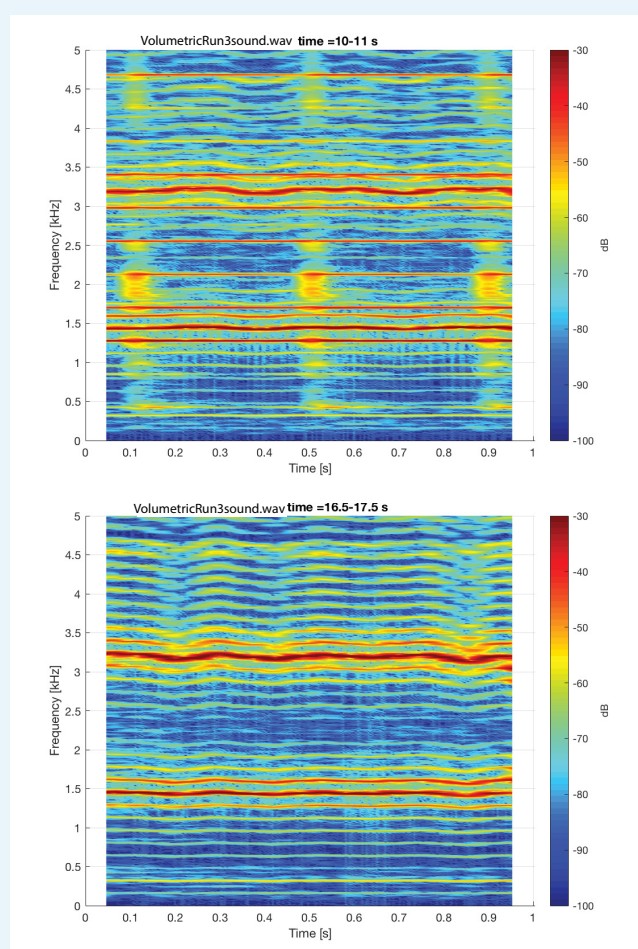

**Appendix 1—figure 19.** Spectrogram of steady-state overtone voicing assocaited with the volumetric scan shown in *Appendix 1—figure 18*. Two different one-second segments are shown: the top segment shows images there were made during the scan (and thus includes acoustic noise from the scanner during image acquisition), while the bottom segment shows images made just after scan ends but while the subject continues to sing.

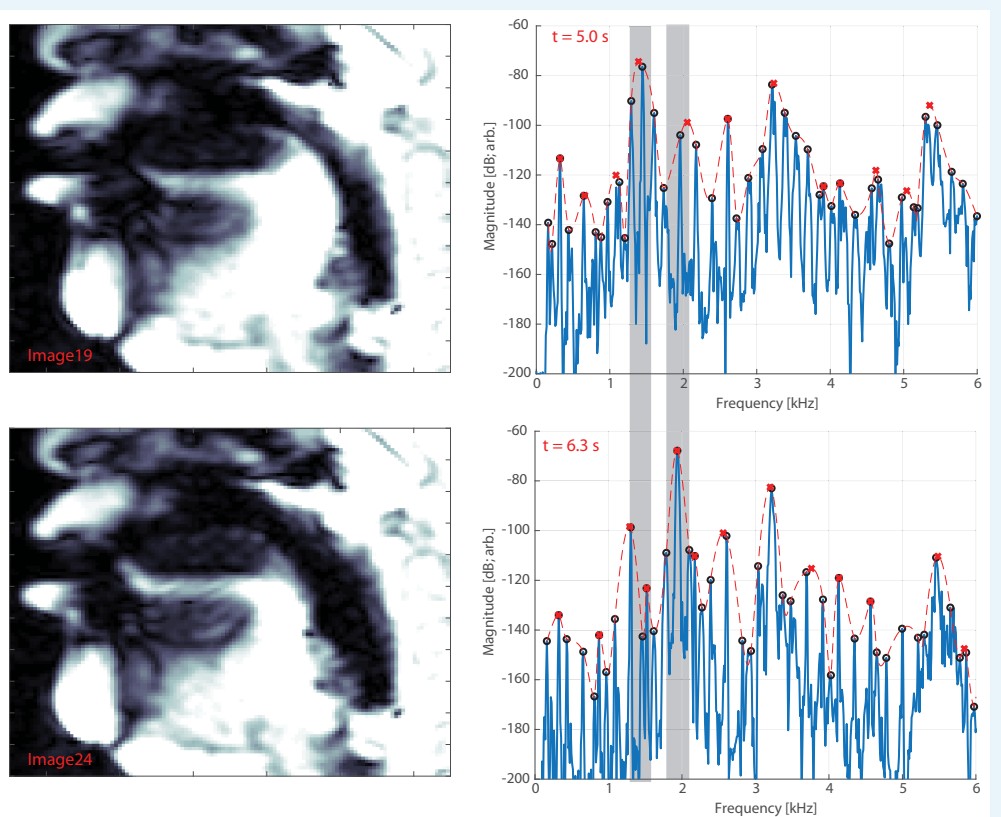

**Appendix 1—figure 20.** Representative movie frames and their corresponding spectra for singer T2, as input into modeling parameters (e.g., *Figure 5*). The corresponding Appendix data files are DynamicRun2S.mov (MRI images) and DynamicRun2sound.wav (spectra; see also DynamicRun2SGrid.pdf). The top row shows a 'low pitch' (first) focused state at about 1.3 kHz whereas the bottom row shows a 'high' pitch at approximately 1.9 kHz. Note a key change is that the back of the tongue moves forward to shift from the low to the high pitch. Thin gray bars are added to the spectra to help to highlight the frequency difference. The legend is the same as that shown in *Figure 1*.

## Data

All data relevant to the study have been placed in the online repository – https://datadryad. org/stash (*Bergevin, 2020*). Below is a list of the data placed there, along with a brief description (see 'Materials and methods' section for additional details).

## Acoustic data

All waveforms were obtained at a sample rate of 96 kHz and a bit-depth of 24 bits.
- T1_1.wav
- T1_2.wav
- T1_3.wav
- T1_3short.wav
- T2_1.wav
- T2_1shortA.wav
- T2_1shortB.wav
- T2_1shortC.wav
- T2_2.wav
- T2_2short.wav
- T2_3.wav
- T2_4.wav

- T2_5.wav
- T2_5longer.wav
- T2_5short.wav
- T3_2.wav
- T3_2shortA.wav
- T3_2shortB.wav
- T4_1.wav
- T4_1shortA.wav

## MRI data

* Images Images were only obtained from singer T2. Note that all image data are saved as DICOM files (i.e., .dcm) :
- Volumetric Run1
- Volumetric Run2
- Volumetric Run3
- Dynamic midsagittal Run1
- Dynamic midsagittal Run2
- Dynamic midsagittal Run3

* Audio recordings acquired during MRI acquisition (see 'Materials and methods').
- Vol. Run1 audio
- Vol. Run2 audio
- Vol. Run3 audio
- Dyn. Run1 audio
- Dyn. Run2 audio
- Dyn. Run3 audio

* MRI Midsagittal movies with sound were also created by animating the frames in Matlab and syncing the recorded audio via Wondershare Filmora. They are saved as .mov files (Apple QuickTime Movie files):
- Dyn. Run1 video
- Dyn. Run2 video
- Dyn. Run3 video

To facilitate connecting movie frames back to the associated sound produced by singer T2 at that moment, the movies include frame numbers. Those have been labeled on the corresponding time location in the spectrograms (see red labels at top):
- Dyn. Run1 spectrogram
- Dyn. Run2 spectrogram
- Dyn. Run3 spectrogram

* Segmented volumetric data files (like those shown in *Figure 3*), data saved as STL files (i. e., .stl):
- Segmented data (T2)

## Software and synthesized song

Simulations and waveform analysis were implemented in Matlab. The TubeTalker software is provided 'as is':
- Code to analyze general aspects of the waveforms (e.g., *Figure 1* spectrograms)
- Code to quantify $e_R$ time course (e.g., *Appendix 1—figure 2*)
- TubeTalker (zipped file, 7 MB)

