## [Decision Letter]

**Acceptance summary:**

Tuvan throat singing, in which people are able to simultaneously produce and independently control two different distinct pitches using the human vocal apparatus, has fascinated hearing and speech researchers for decades. This careful study examines the acoustics of the produced sound and offers new insights into why the produced sound results in two distinct, separately controllable pitches.

**Decision letter after peer review:**

Thank you for submitting your article "Overtone focusing in biphonic Tuvan throat singing" for consideration by *eLife*. Your article has been reviewed by two peer reviewers, one of whom is a member of our Board of Reviewing Editors, and the evaluation has been overseen by Barbara Shinn-Cunningham as the Senior Editor. The reviewers have opted to remain anonymous.

The reviewers have discussed the reviews with one another, and the Reviewing Editor has drafted this decision to help you prepare a revised submission.

Summary

We enjoyed this work addressing mechanisms by which throat singers produce dual pitches that assesses the mechanism for this in terms of the ways in which the vocal tract is precisely controlled based on MRI videos. The work mentions other biological examples of dual fundamentals in songbirds for the broad *eLife* audience. One of the issues that came up in discussion was the control for normal vocalisations without biphonation, but I think the authors make a reasonable case that the singers act as their own controls. Basically, the work shows that the dual pitch mechanism is associated with changes in the vocal tract morphology based on two constrictions that merge second and third formants and is associated with what they call a 'focussed state' in which the harmonics at 1.5kHz to 2kHz are accentuated. The idea as I understand it is that this accentuates a single harmonic of the fundamental glottal pulse rate so that a new high frequency component of Khoomei emerges that is in effect perceptually 'released' from the harmonic series to allow the emergence of the high pitched whistling part of the song.

Major comments

1) From first principles, dual pitch singing could be achieved by a different type of glottal pulse generation in the larynx so that two vibration modes were present (as in avian syrinx). This is not the mechanism suggested here, and it is hard to see how the anatomy and physiology of the human larynx might allow this, but this has not been directly examined in the MRI work. The authors carried out a careful acoustic analysis which shows only one harmonic series before and after transitions to throat singling (without shifting), which I think is adequate. But they might comment on the other possible mechanism for biological readers, if only to dismiss it.

2) Both reviewers thought the discussion of the basis for the perceived dual pitch was not clear. The authors discuss differences in cochlear mechanisms between low frequency regions and high frequency regions. More effort could be made to explain how the dual pitch, which is attributed to a type of spectral emphasis, can be reconciled with current models of pitch perception. The fundamental for the singers assessed was ~150Hz so that the >1.5 kHz region will be unresolved (H10 and above). The greatest contribution to the salience of the low pitch will be the resolved harmonics at frequencies below the focus region, which are well represented. The high-frequency harmonics will usually contribute (weakly) to the low pitch based on the temporal firing patterns due to merged harmonics in frequency bands. The authors appear to be arguing that a different spectral pitch emerges in the high frequency focussed region, distinct from that associated with the lower harmonics.

3) The argument about decreased phase locking at high frequency was not convincing: this occurs in a much higher frequency region that the focussed region. The argument that the high pitch was not easily explained by a non-linear distortion was convincing.

4) In conclusion, we thought the work nicely shows the changes in vocal tract morphology and associated spectrum as an explanation for the dual pitch, but more teasing out of mechanism for the dual pitch perception is required in a way that might be accessible to readers.

[Editors' note: further revisions were suggested prior to acceptance, as described below.]

Thank you for submitting your article "Overtone focusing in biphonic Tuvan throat singing" for consideration by *eLife*. Your article has been reviewed by a member of our Board of Reviewing Editors and Barbara Shinn-Cunningham as the Senior Editor.

We are afraid we are still not satisfied with the discussion of the basis for the dual pitch at the end of the Discussion. The authors have demonstrated a region of spectral focus as a proposed mechanism for the new pitch. But we still do not understand why the focussed overtones produce a different pitch. They are still harmonics of the same fundamental and interactions between them in this unresolved region would be expected to produce beating at the same frequency as the fundamental, in the absence of non-linear mechanisms. We also do not understand what the additional relevance of a decrease in phase locking in this region would be that the authors highlight. Are the authors claiming that the focused region produces spectral excitation in a region without the usual coding of beating between harmonics (because of decreased phase locking) and that this is the cause of the new pitch? If so an explicit suggestion along those lines might help readers who are familiar with conventional pitch models.

*eLife* does not usually encourage multiple rounds of revision but this is a critical point in the interpretation of an interesting study, and I would encourage a revision with a much shorter final section of Discussion that explains a clear hypothesis related to the cause of the new pitch.

---

## [Author Response]

Major comments1) From first principles, dual pitch singing could be achieved by a different type of glottal pulse generation in the larynx so that two vibration modes were present (as in avian syrinx). This is not the mechanism suggested here, and it is hard to see how the anatomy and physiology of the human larynx might allow this, but this has not been directly examined in the MRI work. The authors carried out a careful acoustic analysis which shows only one harmonic series before and after transitions to throat singling (without shifting), which I think is adequate. But they might comment on the other possible mechanism for biological readers, if only to dismiss it.

We attempted to further clarify the point (that we saw no evidence for a nonlinear source mechanism) and added an additional line of text as per the suggestion.

2) Both reviewers thought the discussion of the basis for the perceived dual pitch was not clear. The authors discuss differences in cochlear mechanisms between low frequency regions and high frequency regions. More effort could be made to explain how the dual pitch, which is attributed to a type of spectral emphasis, can be reconciled with current models of pitch perception. The fundamental for the singers assessed was ~150Hz so that the >1.5 kHz region will be unresolved (H10 and above). The greatest contribution to the salience of the low pitch will be the resolved harmonics at frequencies below the focus region, which are well represented. The high-frequency harmonics will usually contribute (weakly) to the low pitch based on the temporal firing patterns due to merged harmonics in frequency bands. The authors appear to be arguing that a different spectral pitch emerges in the high frequency focussed region, distinct from that associated with the lower harmonics.

This criticism was given particularly serious thought and consideration. As a result, we totally rewrote this section to make the proposed ideas clearer, as well as accessible to a broad readership. We tried to find a better balance between issues/questions related to pitch coding and those to cochlear mechanics.

3) The argument about decreased phase locking at high frequency was not convincing: this occurs in a much higher frequency region that the focussed region. The argument that the high pitch was not easily explained by a non-linear distortion was convincing.

As alluded to in the comments above, we clarified the nature of the argument (re phase locking) by expanding upon the discussion of pitch coding. While a reasonable degree of phase locking would still be expected around the 1.5-2 kHz region, this is also where temporal coding starts to fall off dramatically (e.g., Verschooten et al., 2018, PLoS Biol.). That facet, that in the 1-2 kHz region of the human cochlea the fidelity of timing information changes, is what is relevant to the narrative thread here.

4) In conclusion, we thought the work nicely shows the changes in vocal tract morphology and associated spectrum as an explanation for the dual pitch, but more teasing out of mechanism for the dual pitch perception is required in a way that might be accessible to readers.

See comments above.

[Editors' note: further revisions were suggested prior to acceptance, as described below.]

[…]eLife does not usually encourage multiple rounds of revision but this is a critical point in the interpretation of an interesting study, and I would encourage a revision with a much shorter final section of Discussion that explains a clear hypothesis related to the cause of the new pitch.

As we would like to see this work published with *eLife*, we have drastically truncated the highlighted section to create “a much shorter final section” as suggested. Given our lack of expertise in pitch perception, coupled with our appreciation for the comments raised, we instead (succinctly) reframed through the lens of looking ahead at future work. Specifically, we include only what we think are some quite interesting and provocative parallels we have observed between Sygyt song and cochlear mechanics. We believe that providing this as a summary to the narrative will help stimulate crosstalk between emerging viewpoints in cochlear mechanics and central processing (e.g., pitch perception).

As such, hopefully we have a more streamlined “story” that will be sufficient for

publication. We believe the rest of the work paints a clear picture as to how the

morphology leads to biphonation and that can stand on its own without over speculation on other facets.